# Bridging Implicit-Explicit Representations for Ultra-Low Bitrate Image Compression

## Abstract

While recent VAE-based neural codecs achieve impressive results at low bitrates when optimized for perceptual quality, their performance degrades significantly under ultra-low bitrate conditions. To address this, generative methods that exploit semantic priors from pretrained models have emerged, revolutionizing ultra-low bitrate compression. However, these approaches remain constrained by a fundamental tradeoff between semantic faithfulness and perceptual realism. Methods relying on explicit semantic guidance preserve content accuracy but often lack textural fidelity, while those based on implicit representation can generate convincing details but may suffer from semantic drift. In this work, we introduce a unified framework that bridges this gap by coherently integrating explicit and implicit semantic representations. We condition a diffusion model with explicit high-level semantics while using reverse-channel coding to implicitly encode fine-grained information. In addition, a novel plugin encoder provides flexible control over the distortion-perception balance. Extensive experiments demonstrate that our framework achieves state-of-the-art rate–perception performance, outperforming existing approaches and surpassing DiffC by 23.49%, 12.25%, and 23.09% DISTS-BD-Rate on the Kodak, DIV2K, and CLIC2020 datasets, respectively.

## 1 Introduction

The rapid surge in digital visual data has substantially increased the cost of storage and transmission, driving the need for compression techniques that deliver both high compression ratios and high visual quality. At low bitrates (< 0.1 bpp), traditional codecs (Bellard, 2018; Bross et al., 2021) and distortion-oriented VAE-based codecs (He et al., 2022; Jiang & Wang, 2023; Han et al., 2024; Li et al., 2024) tend to over-smooth images, resulting in blurred edges and loss of fine structures. Perceptual optimization (Mentzer et al., 2020; Muckley et al., 2023) alleviates this effect but cannot fully prevent severe artifacts at ultra-low bitrates (< 0.02 bpp).

To overcome over-smoothing, explicit-representation approaches inject pretrained semantic cues into the compression pipeline. GLC (Jia et al., 2024) employs a pretrained VQGAN (Esser et al., 2021) to extract compact latents that are transform-coded, achieving improved perceptual fidelity, yet it remains outside the ultra-low bitrate regime while relying solely on a relatively weak GAN prior (Rombach et al., 2022; Tian et al., 2024). DiffEIC (Li et al., 2025c) introduces stronger diffusion priors by conditioning a diffusion model on compressed features, while PerCo (Careil et al., 2024) combines VQGAN-based latent extraction

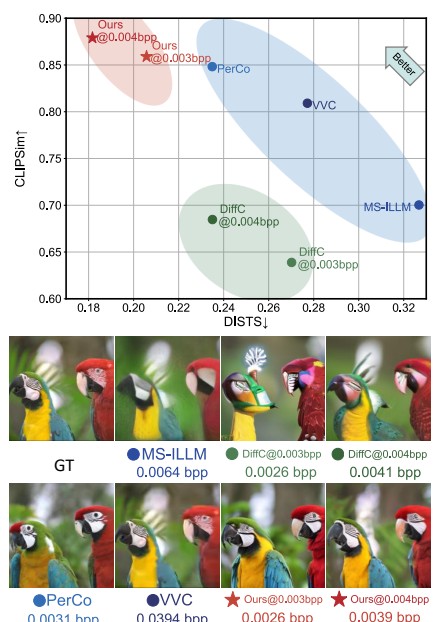

Figure 1: Comparison of different methods at ultra-low bitrates. Blue denotes explicit compression methods, green denotes implicit ones, and red represents our dual representations approach. Our method better preserves textural and semantic details.

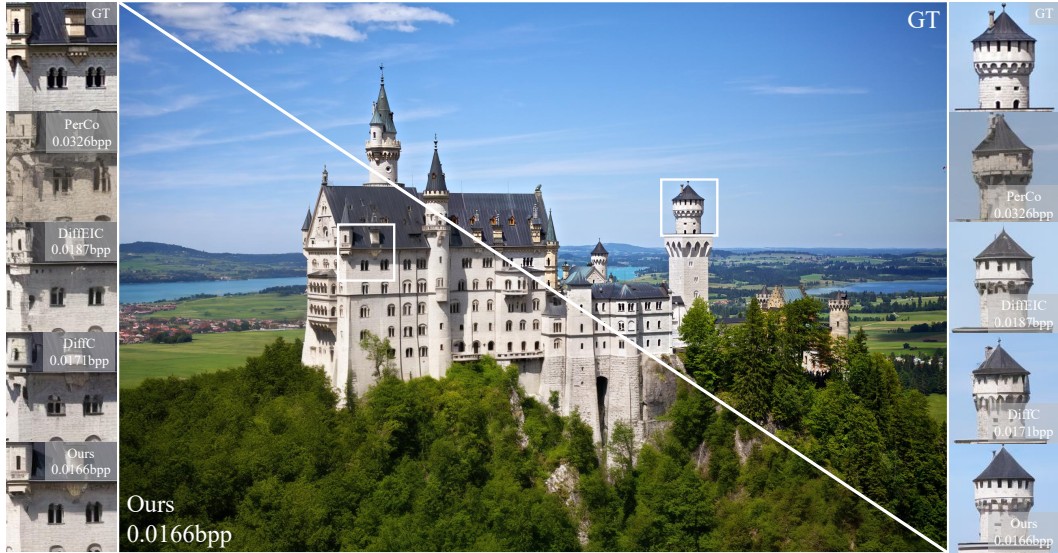

Figure 2: Visual examples and comparisons on 2K-resolution image at ultra-low bitrates. Our method reconstructs more realistic and consistent details with fewer bits. In contrast, PerCo (Careil et al., 2024), DiffEIC (Li et al., 2025c) and DiffC (Vonderfecht & Liu, 2025) exhibit inconsistent details compared to the original images. Best viewed on screen for details.

with diffusion-based refinement. However, most explicit methods apply the generative priors only during decoding, leaving their potential at the encoding stage largely untapped and limiting reconstruction quality at extremely low rates.

In contrast, implicit diffusion-based compression adopts a different paradigm. DiffC (Theis et al., 2022; Vonderfecht & Liu, 2025) leverages a pretrained diffusion model for both encoding and decoding. Instead of compressing explicit semantic features, it compresses noisy samples along the diffusion trajectory, implicitly embedding semantic cues in the parameters of Gaussian noise distributions. Reverse-channel coding (RCC) (Theis & Ahmed, 2022) enables efficient encoding of these noisy latents, supporting bitrates as low as $1 \times 10^{-3}$ bpp. However, due to the inherently incomplete and structurally incoherent semantic information in early noisy steps, DiffC still struggles to produce perceptually satisfactory results at ultra-low bitrates.

These limitations reflect an inherent perception–semantic tradeoff in ultra-low bitrate compression. As shown in Figure 1, explicit approaches (Bross et al., 2021; Careil et al., 2024; Li et al., 2025c) preserve global content but under-render high-frequency details, whereas implicit methods (Vonderfecht & Liu, 2025) generate rich textures without stable semantic anchors and are prone to drift, especially in early noisy stages.

To overcome the limitations of existing methods, we introduce a dual semantic compression framework that bridges explicit and implicit representations. Our framework is compatible with various image compression pipelines employing conditional diffusion models. By jointly leveraging explicit and implicit semantics, it mitigates the semantic degradation observed in DiffC (Vonderfecht & Liu, 2025) at ultra-low bitrates and enhances reconstruction fidelity in approaches relying solely on explicit information (Careil et al., 2024; Li et al., 2025c). Specifically, we first encode explicit semantic content (e.g., text and latent representations), which is then complemented by implicit semantics derived from diffusion steps via RCC. At extremely low bitrates, traditional caption-style prompts can dominate the bit budget. To improve efficiency, we replace them with compact, tag-style prompts and compress them with a more bit-wise design. Furthermore, to tackle pixel-level fidelity degradation at low bitrates, we introduce a plugin module that modulates the target features used for implicit compression, enabling controllable tradeoffs between distortion and perceptual quality without modifying the existing architecture. Additionally, to improve quality for high-resolution inputs, we incorporate a tile-based inference strategy that ensures global consistency while enhancing reconstruction quality. Our contributions can be summarized as follows:

- We propose a dual semantic compression framework that jointly exploits explicit and implicit semantics, achieving high perceptual quality at ultra-low bitrates.

- We design a compact semantic extractor using tag-style prompts to reduce bitrate overhead, and a plugin module for controllable distortion-perception tradeoffs.

- Extensive experiments show that our method consistently surpasses state-of-the-art approaches at bitrates as low as $3 \times 10^{-3}$ bpp, delivering superior performance and surpassing DiffC by 44.86% and 35.75% CLIPSim-BD-rate on the Kodak and DIV2K datasets.

## 2 RELATED WORK

### 2.1 PERCEPTUAL IMAGE COMPRESSION

Blau & Michaeli (2019) formally reveal a fundamental tradeoff among rate, distortion, and perception, suggesting that optimizing all three objectives simultaneously is inherently constrained. Consequently, recent research (Lu et al., 2024; Ballé et al., 2025; Liang et al., 2025; Relic et al., 2025) has increasingly focused on improving perceptual quality by allowing imperceptible distortions. HiFiC (Mentzer et al., 2020) introduces a generative adversarial network (GAN) (Goodfellow et al., 2014) to improve visual fidelity, while MS-ILLM (Muckley et al., 2023) replaces the discriminator with a non-binary one. CDC (Yang & Mandt, 2023) conditions a diffusion-based decoder on compressed features. DiffC (Theis et al., 2022) compresses the noise-corrupted pixels using RCC and then reconstructs via diffusion. Control-GIC (Li et al., 2025a) extracts multi-scale features using VQGAN and compresses the feature indexes using entropy coding. Despite these advances, these methods struggle at ultra-low bitrates due to insufficient exploitation of semantic priors.

To address the distortion introduced by perceptual optimization, several methods have explored controllable perception-distortion tradeoffs. Zhang et al. (2021) and Yan et al. (2022) propose using different decoders for reconstructions optimized for distortion and perception, respectively. MRIC (Agustsson et al., 2023) introduces a tunable hyperparameter to balance distortion and perceptual losses. DIRAC (Ghouse et al., 2023) modulates the amount of added detail by controlling the number of diffusion steps.

### 2.2 ULTRA-LOW BITRATE IMAGE COMPRESSION

By leveraging powerful generative priors, ultra-low bitrate compression becomes feasible by encoding only minimal explicit semantic information (Zhang et al., 2025a; Xue et al., 2025). GLC (Jia et al., 2024) extracts low-dimensional features using VQGAN and further compresses them via transform coding, achieving visually appealing results at low bitrates. More approaches have primarily adopted pretrained diffusion models. Text+Sketch (Lei et al., 2023) conditions latent diffusion models (LDMs) (Rombach et al., 2022) on textual descriptions and edge information to produce semantically aligned reconstructions. PerCo (Careil et al., 2024) vector-quantizes VAE features, encodes the resulting indexes into a bitstream, and uses them as conditioning signals for guided diffusion. DiffEIC (Li et al., 2025c) builds upon a pretrained LDM and is trained end-to-end to compress control information that guides the generative process.

Most existing methods, however, exploit generative priors only at the decoding stage, limiting semantic preservation. Vonderfecht & Liu (2025), an extension of DiffC (Theis et al., 2022), employs text-to-image diffusion models for both encoding and decoding but its purely implicit scheme often suffers from semantic misalignment at ultra-low bitrate constraints. These limitations motivate our dual representations framework, which integrates explicit and implicit cues to enable faithful and realistic reconstructions.

## 3 PRELIMINARY

### 3.1 DIFFUSION DENOISING PROBABILISTIC MODELS

DDPMs (Ho et al., 2020) generate data through a sequence of iterative, stochastic denoising steps. The joint distribution of the data $x_0$ and latent variable $x_{1:T}$ is learned through the model, i.e., $p_\theta(x_0) = \int p_\theta(x_0, x_{1:T})dx_{1:T}$. The diffusion process $q$ gradually corrupts the data with noise,

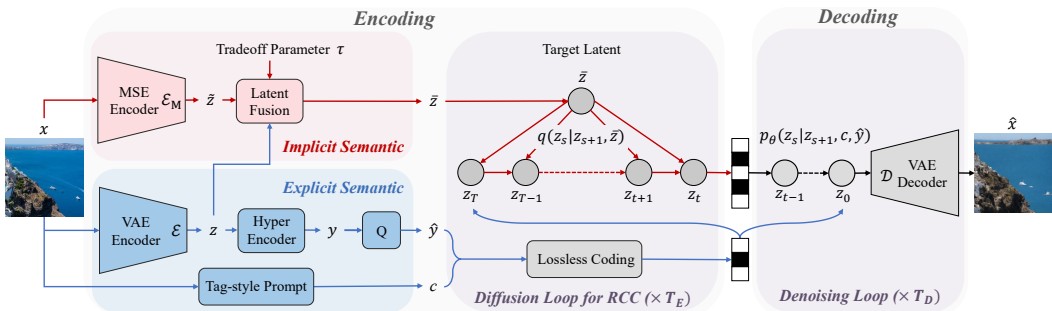

Figure 3: Overview of the proposed dual semantic compression framework. Explicit semantics consist of the quantized latent $\hat{y}$ and a tag-style text prompt $c$, while implicit semantics are derived from noise-corrupted latents using RCC.

while the reverse process $p_\theta$ reconstructs the structure. Both processes follow Markov dynamics, controlled by a monotonically increasing sequence of noise variances $\beta_t \in (0,1)$. Specifically, the forward process is defined as $q(x_t|x_{t-1}) = \mathcal{N}(x_t|\sqrt{\alpha_t}x_{t-1}, \beta_t\mathbf{I})$, where $\alpha_t = 1 - \beta_t$ and $x_t$ can be directly sampled from $x_0$ using $x_t(x_0) = \sqrt{\bar{\alpha}_t}x_0 + \sqrt{1 - \bar{\alpha}_t}\epsilon$, where $\bar{\alpha}_t = \prod_{s=1}^{t}\alpha_s$. The reverse process is approximated as $p_\theta(x_{t-1}|x_t) = \mathcal{N}(x_{t-1}|\epsilon_\theta(x_t, t), \beta_t\mathbf{I})$, where $\epsilon_\theta(x_t, t)$ is trained to predict the noise $\epsilon$ added to $x_0$ at timestep $t$.

During inference, denoised image at each step is obtained by subtracting the predicted noise from the noisy input:

$$x_{t-1} = \frac{1}{\sqrt{\alpha_t}}\left(x_t - \frac{\beta_t}{\sqrt{1 - \bar{\alpha}_t}}\epsilon_\theta(x_t, t)\right) + \sigma_t\epsilon, \tag{1}$$

where $\sigma_t$ controls the level of stochasticity during sampling.

### 3.2 Reverse-Channel Coding

Reverse-channel coding (RCC) (Theis & Ahmed, 2022) addresses a fundamental problem in information theory by enabling efficient communication of a random variable $x \sim q(x)$ through a shared reference distribution $p(x)$ accessible to both the encoder and the decoder. It aims to encode $x$ such that the expected number of transmitted bits approximates the Kullback-Leibler (KL) divergence $D_{KL}(q||p)$, effectively exploiting the statistical similarity between $q$ and $p$. In the context of image compression, DiffC (Theis et al., 2022) adopts RCC to compress noisy variables $x_t$ sampled from the posterior distribution $q(x_t|x_{t+1}, x_0)$ along the diffusion trajectory, using the reverse diffusion distribution $p_\theta(x_t|x_{t+1})$ as the reference. During decoding, the decoder reconstructs $q(x_t|x_{t+1}, x_0)$ based on $p_\theta(x_t|x_{t+1})$ to resample $x_t$, and the denoising process continues until $x_0$ is recovered.

The Poisson Functional Representation algorithm (Theis & Ahmed, 2022) enables RCC but suffers from exponential runtime when $D_{KL}(q||p)$ is large and reduced bit efficiency when it is too small. To overcome these limitations, Vonderfecht & Liu (2025) propose skipping denoising steps whose per-step $D_{KL}$ is negligible, and decomposing the overall distributions pair $(q,p)$ into approximately independent components $(q_i, p_i)$, such that each $D_{KL}(q_i||p_i)$ remains in an efficient operational range. These strategies significantly enhance both scalability and bit efficiency of RCC-based framework.

## 4 Proposed Method

In this section, we present the overall framework of the proposed method. As illustrated in Figure 3, an input image $x$ is first encoded into a VAE latent $z = \mathcal{E}(x)$. A hyper encoder then produces an explicit latent representation $y$, which is quantized and entropy-coded into $\hat{y}$. In parallel, an image-to-tags module extracts a compact set of tag indexes $c \in \{1, \ldots, N\}^K$ to capture high-level semantics. To enhance pixel fidelity at ultra-low bitrates, a distortion-oriented encoder $\mathcal{E}_M$ generates $\tilde{z} = \mathcal{E}_M(x)$. A distortion–perception knob $\tau \in [0, 1]$ blends the two latents as $\bar{z} = \tau z + (1 - \tau)\tilde{z}$,

which serves as the compression target for RCC along a conditional diffusion trajectory guided by $(c, \hat{y})$. Joint conditioning on explicit representations and implicitly transmitted diffusion states fuses high-level semantics with fine-grained details for more faithful reconstructions. The bitstream consists of three parts: the latent $\hat{y}$, the tag codes for $c$, and RCC codes for noisy states $z_{T:T-T_E}$. At the decoder, $(c, \hat{y})$ are first decoded, RCC is inverted to recover $z_{T:T-T_E}$, and conditional denoising is performed from $z_{T-T_E}$ for $T_D$ steps to obtain a clean latent $\hat{z}$. The final reconstruction is $\hat{x} = \mathcal{D}(\hat{z})$. RCC steps $T_E$ and the knob $\tau$ provide continuous control of the rate–distortion–perception tradeoff without altering the conditional diffusion architecture, while a tile-wise strategy ensures stable processing of high-resolution inputs.

## 4.1 EXPLICIT SEMANTIC INFORMATION

We use explicit semantic cues to anchor high-level content and stabilize conditional generation under ultra-low bitrates. The explicit part provides two signals: a quantized latent $\hat{y}$ that captures coarse appearance and geometry, and a compact set of tag indexes $c$ that encodes high-level semantics.

For latent representation, an input image $x$ is first encoded by a pretrained LDM encoder $\mathcal{E}$ to obtain a latent $z$. To enable efficient quantization and compression, $z$ is further transformed by a hyper encoder into a compact latent $y$, which is then quantized, typically via vector or scalar quantization, to produce $\hat{y}$. The quantized latent $\hat{y}$ is subsequently entropy-encoded into a binary bitstream.

For prompts, existing methods typically rely on caption-style descriptions generated by BLIP (Li et al., 2022). However, such captions are often verbose, contain redundant qualifiers, and poorly align with fine-grained image semantics, causing unnecessary bitrate overhead (Wu et al., 2024). Instead, we adopt tag-style prompts generated by RAM (Zhang et al., 2024), where each token represents a distinct visual concept. This representation eliminates redundancy and provides a more compact extraction of essential semantics. To exploit the limited vocabulary and structured redundancy of tag-style prompts, we encode each tag with fixed-length codes of $\lceil \log_2 N \rceil$ bits, where $N$ is the vocabulary size, enabling more efficient bitrate allocation. For a prompt $c \in \{1, \ldots, N\}^K$ containing $K$ tags, the total cost is $K \lceil \log_2 N \rceil$ bits. Serving both as a base reconstruction cue and a conditioning signal for the diffusion prior, $(\hat{y}, c)$ improves alignment between the generative trajectory and the target distribution.

## 4.2 IMPLICIT SEMANTIC INFORMATION

While explicit semantics capture coarse structures, fine-grained textures and stochastic details remain challenging to transmit under extreme compression budgets. To address this, we adopt an implicit representation based on diffusion priors. We compress intermediate noisy states along the diffusion trajectory using RCC (Theis & Ahmed, 2022), but crucially incorporate explicit conditioning $(c, \hat{y})$ into the prior distribution. Formally, for timestep $t$, the encoder samples a noisy latent $z_t \sim q(z_t|z_{t+1}, z_0)$ and encodes it with respect to the conditional distribution $p_\theta(z_t|z_{t+1}, c, \hat{y})$. The expected bitrate of step $t$ approximates the Kullback-Leibler divergence $D_{KL}(q(z_t|z_{t+1}, z_0)||p_\theta(z_t|z_{t+1}, c, \hat{y}))$ via RCC, and the total implicit bitrate across the encoded timesteps is given in Equation 2. Increasing the number of encoded timesteps (larger $T_E$) transmits more information, leading to a higher implicit bitrate and improved reconstruction fidelity.

$$\text{bits} = \sum_{i=T}^{T-T_E} D_{KL}(q(z_i|z_{i+1}, z_0)||p_\theta(z_i|z_{i+1}, c, \hat{y})). \quad (2)$$

## 4.3 DECODING WITH CONDITIONAL DIFFUSION

The distribution $p_\theta(z_t|z_{t+1}, c, \hat{y})$ is shared by both the encoder and decoder, allowing the decoder to first reconstruct the posterior $q(z_t|z_{t+1}, z_0)$ from the compressed bitstream and then sample $z_t$. This sampled latent $z_t$ serves as the starting point for the subsequent reverse diffusion process, which is conditioned on the explicit semantics $(c, \hat{y})$ via mechanisms such as cross-attention or Control-Net (Zhang et al., 2023). Through iterative denoising, the model recovers a noise-free latent representation $\hat{z}$, which is then passed through the VAE decoder $\mathcal{D}$ to produce the final reconstructed image $\hat{x}$. Notably, a higher bitrate, achieved by encoding more diffusion steps $T_E$, allows the de-

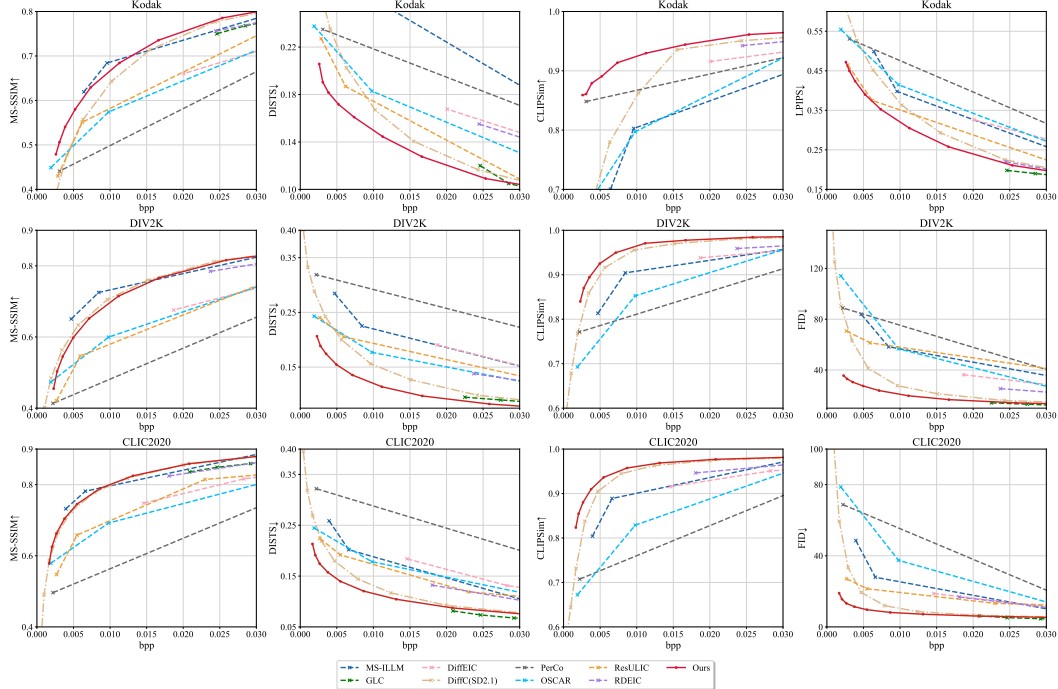

Figure 4: Rate–metric curves on Kodak, DIV2K, and CLIC2020 datasets. Arrows indicate whether higher (↑) or lower (↓) values are better. See supplementary for more results.

coder to initiate the reverse process from a later stage, reducing the number of additional denoising steps $T_D$ required during reconstruction.

### 4.4 DISTORTION-PERCEPTION TRADEOFF

The reconstruction fidelity of LDM is fundamentally constrained by the capacity of the underlying VAE. This limitation becomes more pronounced under low bitrate conditions. To address this issue, we propose improving pixel-level fidelity by modifying the transmission target used in RCC, specifically the reference latent $z_0$ in the posterior $q(z_t|z_{t+1}, z_0)$. Instead of altering the decoder, we introduce a plugin module that adjusts the encoding target, enabling fidelity enhancement while maintaining compatibility with the original coding process.

Formally, let the standard VAE encoding and decoding processes be represented as $z = \mathcal{E}(x)$ and $\hat{x} = \mathcal{D}(z)$. We keep the decoder $\mathcal{D}$ fixed and introduce an auxiliary encoder $\mathcal{E}_M$, which is optimized specifically for pixel-wise reconstruction accuracy. $\mathcal{E}_M$ is trained by minimizing the MSE between the original image and the decoded output, using the loss function $\mathcal{L} = ||x - \mathcal{D}(\mathcal{E}_M(x))||_2^2$. During inference, we enable a continuous tradeoff between perception and distortion by introducing a mixing coefficient $\tau \in [0, 1]$. A fused latent representation $\bar{z}$ is computed by interpolating between the original latent $z$ and the distortion-oriented latent $\tilde{z} = \mathcal{E}_M(x)$, as defined by $\bar{z} = \tau \times z + (1 - \tau) \times \tilde{z}$. The blending latent $\bar{z}$ is then used as the RCC target $z_0$, allowing for controllable distortion-perception tradeoffs without any changes to the decoder.

### 4.5 TILE-BASED PROCESSING

When scaling to high-resolution inputs, standard diffusion inference often encounters memory limitations and produces unrealistic textures (Jiménez, 2023; Wang et al., 2024b; Vonderfecht & Liu, 2025). To address these, we utilize a tile-based strategy in the latent space. The noisy latent is divided into overlapping spatial tiles, and each tile is processed independently and in parallel. To seamlessly merge the outputs of individual tiles, we apply a Gaussian weighting mask to each tile before aggregation. For every pixel in the latent, multiple overlapping predictions are combined and normalized by the sum of their corresponding weights. This weighted blending ensures smooth

Table 1: Detailed BD-Rate(%) of different ultra-low-bitrate image compression models on Kodak, DIV2K and CLIC2020 testset.

| Model | Kodak | | | DIV2K | | | CLIC2020 Test | | |
|---|---|---|---|---|---|---|---|---|---|
| | LPIPS | DISTS | CLIPSim | DISTS | CLIPSim | FID | DISTS | CLIPSim | FID |
| MS-ILLM (ICML'23) | 14.50 | 213.84 | 54.07 | 110.44 | 75.82 | 162.81 | 82.73 | 66.81 | 91.12 |
| GLC (CVPR'24) | -11.85 | 0.39 | - | -5.75 | - | -23.46 | -20.63 | - | -14.01 |
| DiffEIC (TCSVT'24) | 64.18 | 107.06 | 38.89 | 183.96 | 119.42 | 139.85 | 217.89 | 160.41 | 107.77 |
| PerCo (ICLR'24) | 46.68 | 68.97 | -18.10 | 282.32 | 174.80 | 224.93 | 395.80 | 227.85 | 407.13 |
| OSCAR (NeurIPS'25) | 21.26 | 11.26 | 37.32 | 25.39 | 243.94 | 214.42 | 87.75 | 249.86 | 336.84 |
| ResULIC (ICML'25) | -24.70 | -13.13 | - | 37.75 | - | 165.83 | 53.15 | - | 9.15 |
| RDEIC (TCSVT'25) | -3.66 | 93.89 | 8.69 | 81.12 | 66.81 | 64.30 | 78.86 | 67.04 | 94.91 |
| DiffC (ICLR'25) | 0.00 | 0.00 | 0.00 | 0.00 | 0.00 | 0.00 | 0.00 | 0.00 | 0.00 |
| Ours | **-27.11** | **-23.49** | **-48.38** | **-12.25** | **-21.15** | **-53.06** | **-23.09** | **-10.47** | **-61.99** |

transitions at tile boundaries and effectively reduces artifacts caused by discontinuities. The overlapping design not only enhances perceptual quality but also supports efficient parallelization during the diffusion sampling process.

In addition to tiling the noisy latents, we also partition the explicit semantic information $(c, \hat{y})$ accordingly. Since different tiles may contain distinct semantic content, we extract localized prompts for each tile individually, using a tag-based format. To limit the impact on bitrate, we constrain the number of tags per tile. This design maintains a compact textual representation and preserves semantic alignment under extremely low bitrate conditions.

## 5 EXPERIMENTS

### 5.1 SETTINGS

**Training Details for MSE Encoder.** The explicit–implicit integration mechanism in our framework, including the plugin fusion strategy, operates in a fully training-free manner and can be directly applied to any conditional diffusion-based codec. To enhance pixel-level fidelity at low bitrates, we train a plugin distortion-oriented VAE encoder. Specifically, we initialize the encoder with the weights of the original perception-oriented VAE while keeping the decoder frozen. Training is conducted on the high-quality Flickr2W dataset (Liu et al., 2020), where images are randomly cropped to a resolution of $256 \times 256$. We adopt the AdamW optimizer (Loshchilov & Hutter, 2019) with a batch size of 4 and a learning rate of $10^{-5}$.

**Evaluation Metrics.** We evaluate the proposed method on full-resolution images using both distortion and perceptual quality metrics. Bit-per-pixel (bpp) is used to quantify the average number of bits required to encode each pixel. To assess image quality, we report several widely used metrics, including PSNR, LPIPS (Zhang et al., 2018), DISTS (Ding et al., 2022), CLIPSim (Radford et al., 2021), and Fréchet Inception Distance (FID) (Heusel et al., 2017). CLIPSim measures the cosine similarity between CLIP embeddings of the original and reconstructed images (Li et al., 2025b), by resizing the images to $224 \times 224$ to satisfy the input resolution of the pretrained CLIP model while preserving high-level semantic information. And FID is computed by dividing each image into non-overlapping $256 \times 256$ patches (Mentzer et al., 2020; Yang & Mandt, 2023). All evaluations are conducted on three widely used benchmark datasets: the Kodak dataset, the DIV2K test set, and the CLIC2020 test set.

**Comparison Methods.** We compare our method with representative approaches across several categories. These include the GAN-based method MS-ILLM (Muckley et al., 2023); the tokenizer-based method GLC (Jia et al., 2024); diffusion-based explicit compression frameworks such as DiffEIC (Li et al., 2025c), PerCo (Careil et al., 2024), OSCAR (Guo et al., 2025), ResULIC (Ke et al., 2025), and RDEIC (Li et al., 2025d); and the diffusion-based implicit compression framework DiffC (Vonderfecht & Liu, 2025).

### 5.2 MAIN RESULTS

We compare our approach with several state-of-the-art methods, as shown in Figure 4 and Table 1. It shows that our method consistently improves all metrics on Kodak, DIV2K, and CLIC2020 datasets,

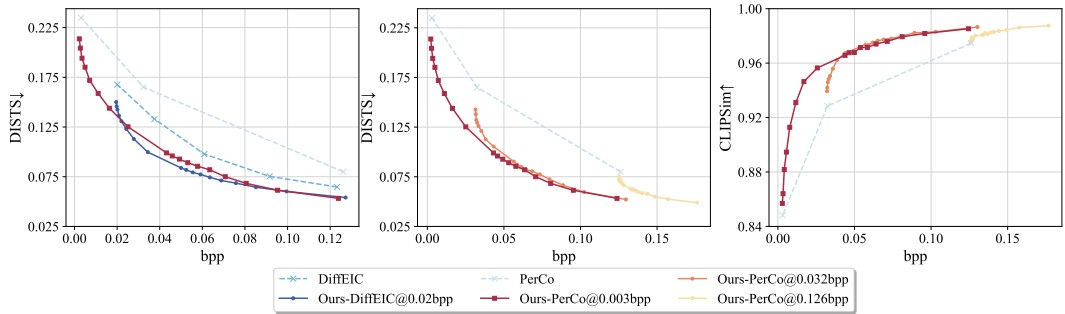

Figure 5: Rate–metric curves on the Kodak dataset. Our method is applied to multiple versions of each base model, trained at different bitrates, resulting in several curves per model. Our method consistently outperforms the corresponding baselines across all bitrate settings.

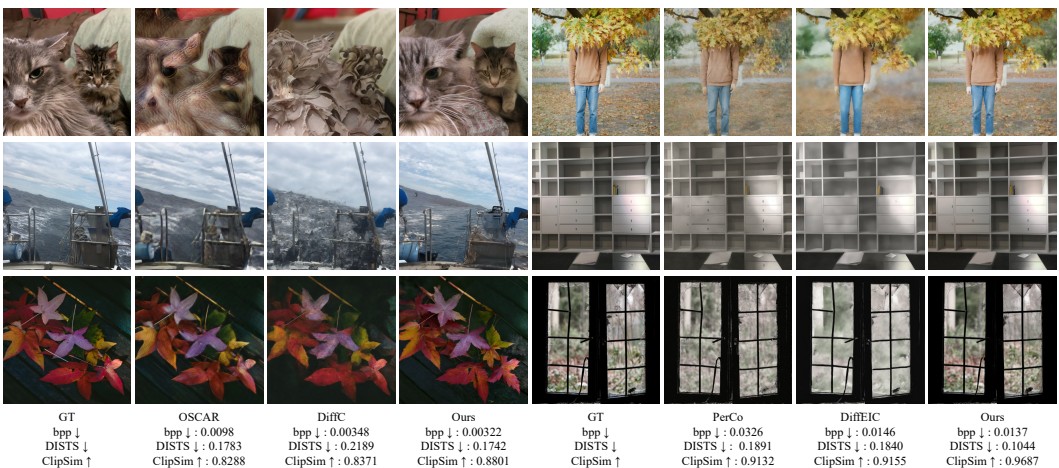

Figure 6: Qualitative comparisons of different methods on CLIC2020 dataset. Our method reconstructs more realistic and consistent details using fewer bits at both ultra-low and low bitrates.

with the largest gains in the $< 0.02$ bpp regime. MS-ILLM (Muckley et al., 2023), which relies on GAN to enhance perceptual quality, produces less visually appealing results. DiffC (Vonderfecht & Liu, 2025), which depends entirely on implicit semantic information, suffers from a sharp drop in performance under ultra-low bitrates. In contrast, our method maintains high perceptual quality by jointly leveraging both types of semantic cues. Specifically, our method achieves a 23.49% bitrate reduction compared to DiffC while maintaining comparable DISTS on the Kodak dataset. Furthermore, at low bitrates, our method delivers higher perceptual quality than GLC (Jia et al., 2024), achieving better DISTS scores on the Kodak dataset.

## 5.3 COMPATIBILITY WITH VARIOUS BASE CODECS

To validate the effectiveness of our framework, we integrate it with two representative baselines for explicit semantics extraction: DiffEIC (Li et al., 2025c) and PerCo (Careil et al., 2024). Our framework is compatible with any conditional diffusion-based compression model trained at arbitrary bitrates. We adopt the feature extraction strategies used in the corresponding baselines for explicit semantics. Specifically, since DiffEIC does not incorporate textual prompts during training, we omit the text prompts $c$ in our variant to align with its original training setup.

As shown in Figure 5, our dual semantic compression framework achieves consistent performance gains under different base models and bitrate levels. Unlike DiffEIC and PerCo, which rely solely on explicit semantics, our approach integrates both explicit and implicit semantic information, resulting in significant improvements in perceptual quality. For example, our PerCo-based variant achieves a 75.97% bitrate reduction over PerCo while maintaining comparable LPIPS performance.

Table 2: BD-Rate (%) comparison of different strategies on Kodak dataset. "FL" denotes fixed-length encoding. Weight $w$ controls the weight of classifier-free guidance. Lower BD-Rate indicates better performance.

| BD-Rate (%) ↓ | Representation | | | | Prompt | | | Guidance (Varying $w$) | | |
|---|---|---|---|---|---|---|---|---|---|---|
| | Explicit | Implicit | Implicit+$c$ | Dual | Caption (zlib) | Tag (zlib) | Tag (FL) | $w=2$ | $w=3$ | $w=4$ |
| **DISTS** | +149.99 | 0.00 | -8.84 | **-23.49** | -11.62 | -12.13 | **-23.49** | -20.22 | **-23.49** | -17.35 |
| **CLIPSim** | +22.92 | 0.00 | -23.62 | **-49.78** | -35.16 | -38.12 | **-49.78** | -44.86 | **-49.78** | -40.80 |

Similarly, our DiffEIC-based variant yields a 37.94% bitrate saving, even though DiffEIC operates in a relatively higher bitrate range. We also observe that our framework yields greater improvements when applied to base models trained at lower bitrates. In such cases, the explicit branch requires only a very small bitrate to act as an effective structural anchor, and this minimal allocation consistently yields the best results, as demonstrated in Appendix F. Based on this finding, we adopt the lowest-bitrate version of PerCo as the default base model in the following comparisons.

## 5.4 QUALITATIVE EVALUATION

Visual results are presented in Figure 6. Our method produces semantically faithful reconstructions with fine details, while competing methods fail to achieve satisfactory results even at higher bpp. For example, MS-ILLM yields severely blurred outputs under ultra-low bitrate conditions, while DiffC introduces semantic inaccuracies and unnatural artifacts. At low bitrates, PerCo and DiffEIC exhibit notable deviations in semantic structure. In comparison, our method maintains strong semantic alignment and finer textures across all bitrate levels.

## 5.5 DISTORTION-PERCEPTION TRADEOFF

By introducing a plugin encoder, we can adjust the compression target of the implicit branch, thereby controlling the final perception–distortion balance. As shown in Figure 7, setting $\tau = 0$ (favoring distortion) yields a 1.62 dB gain in PSNR over $\tau = 1$ (favoring perception) on the Kodak dataset at 0.1229 bpp. Unlike existing tradeoff mechanisms (Agustsson et al., 2023; Ghouse et al., 2023), our approach enables effective adjustment even under ultra-low bitrate conditions and further supports fine-grained controllability across the three-dimensional rate–distortion–perception space, while requiring no modification or retraining of the original compression framework. Additional experiments on the DIV2K dataset are provided in the Appendix G, further confirming the consistency of this controllability.

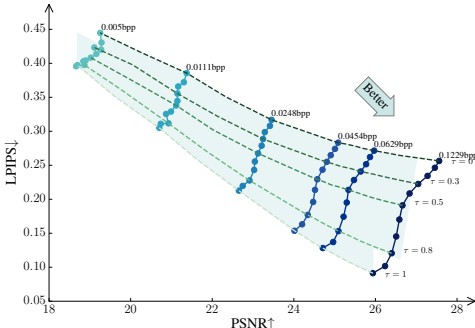

Figure 7: Distortion (PSNR) vs. perception (LPIPS) on Kodak dataset for different rate-distortion-perception tradeoffs. Dashed lines indicate results with the same $\tau$, while solid lines connect results with similar bitrates.

## 5.6 ABLATION STUDY

**Dual Representations.** To validate the effectiveness of our design, we first compare the contributions of different representations in Table 2. Using only explicit or implicit semantics results in suboptimal performance, while combining both representations achieves the best overall results. Specifically, introducing our tag-style caption $c$ into the implicit branch significantly improves semantic consistency, achieving a 23.62% gain in CLIPSim-BD-Rate. Building on this, the complete dual representations framework delivers a significant 49.78% bit saving.

**Prompt Strategies.** We further compare different prompt strategies. Replacing caption-style prompts with tag-style prompts yields consistent improvements, though the full compression potential of tag-based prompts remains untapped. Moreover, switching from Lempel-Ziv coding as implemented in the zlib library (zlib) to our fixed-length encoding scheme enables lossless bitrate re-

duction, decreasing the tag-style prompt bitrate from approximately $2 \times 10^{-3}$ bpp to around $3 \times 10^{-4}$ bpp, resulting in notable overall bitrate savings. These results underscore the advantage of using tag-style prompts in rate-constrained scenarios.

**Classifier-free Guidance.** We evaluate the effect of classifier-free guidance (Ho & Salimans, 2021; Yu et al., 2024). At each diffusion step $t$, $\tilde{\epsilon}_t = \epsilon_t^{\text{pos}} + w \times (\epsilon_t^{\text{pos}} - \epsilon_t^{\text{neg}})$, where $\epsilon_t^{\text{pos}} = \epsilon_\theta^{\text{pos}}(z_t, (\hat{y}, c))$ uses our explicit cues $(\hat{y}, c)$ and $\epsilon_t^{\text{neg}} = \epsilon_\theta^{\text{neg}}(z_t, (\hat{y}, \text{neg}))$ uses common negative prompts neg. We find that setting $w = 3$ yields the best visual quality, which we adopt as the default.

**Partitioning Strategy.** We validate the partitioning strategy in Table 5. Although adding text prompts causes a slight decrease in DISTS, it significantly enhances semantic consistency, yielding 11.18% and 6.45% savings in CLIPSim-BD-Rate and FID-BD-Rate, respectively. These results confirm the effectiveness of the proposed tile-based strategy for memory-efficient, high-fidelity compression. Furthermore, we select a tile size of 512 pixels, as it provides the best perceptual performance, as shown in Appendix H.

Table 3: BD-Rate (%) compression under different partitioning strategies on DIV2K.

| Strategy | w/o Tile + $c$ | Tile + w/o $c$ | Tile + $c$ |
|---|---|---|---|
| DISTS | 0.00 | -38.24 | -37.58 |
| CLIPSim | 0.00 | -24.45 | -35.63 |
| FID | 0.00 | -37.08 | -43.53 |

### 5.7 RUNTIME ANALYSIS

Compared with DiffC (Vonderfecht & Liu, 2025), we replace the standard text-to-image diffusion backbone with a conditional diffusion model designed for image compression. Although conditional diffusion usually requires fewer sampling steps, state-of-the-art codecs such as PerCo, DiffEIC, and DiffC still rely on tens of denoising steps, leading to decoding latencies of 1~10 seconds. Relative to DiffC, our framework incurs additional computation from the ControlNet-style conditioning modules. Compared with PerCo, our method uses

Table 4: Runtime and DISTS-BD-Rate comparison on the Kodak dataset.

| Method | Enc (s) | Dec (s) | BD-Rate (%) |
|---|---|---|---|
| PerCo | 0.39 | 1.92 | 68.97 |
| DiffEIC | 0.23 | 4.82 | 107.06 |
| OSCAR | 0.09 | 0.11 | 11.26 |
| DiffC | 0.6–9.1 | 2.9–8.4 | 0 |
| Ours | 0.73–9.56 | 2.29–9.39 | **-25.02** |

more diffusion steps and additionally incorporates the RCC process. Our experiments show that on the Kodak dataset, adding a very small implicit branch with only three additional diffusion steps on top of the explicit branch slightly increases decoding time from 1.92s to 2.29s and bitrate from 0.002194 to 0.002591 while delivering substantial perceptual gains as DISTS improves from 0.3188 to 0.2172. While our speed does not match one-step diffusion models (Guo et al., 2025; Zhang et al., 2025a; Xue et al., 2025), our method is fully plug-and-play and requires no task-specific training, enabling broad applicability. Although runtime remains suboptimal, techniques such as caching (Ma et al., 2024), quantization (Shang et al., 2023; Wang et al., 2024a), pruning (Fang et al., 2023; Zhang et al., 2025b), and RCC optimizations (Ohayon et al., 2025) present promising directions for reducing computation, though integrating them is beyond this work's scope.

## 6 CONCLUSION

In this work, we propose a dual semantic compression framework that jointly leverages both explicit and implicit representations for ultra-low bitrate image compression. The explicit representations deliver compact prompts and latent features that capture high-level semantics and serve as conditioning signals for the diffusion model, while the implicit representations capture fine-grained visual details through progressive latent refinement along the diffusion trajectory. This collaborative design ensures both semantic consistency and perceptual fidelity under extreme compression. In addition, we introduce a distortion-perception tradeoff module that adjusts the implicit compression target without altering the decoder, enabling flexible quality control at inference time. Extensive experiments demonstrate that our method consistently outperforms existing approaches in both perceptual quality and pixel-level accuracy, especially at ultra-low bitrates.

## ETHICS STATEMENT

All the authors read and adhere to the ICLR Code of Ethics.

## REPRODUCIBILITY STATEMENT

We ensure that all experiments in this paper are fully reproducible as described in the main text.

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

## A    PRETRAINED BASELINES

For MS-ILLM (Muckley et al., 2023), DiffEIC (Li et al., 2025c), and DiffC (Vonderfecht & Liu, 2025) baseline, we use the official codebase [1][2][3]. For PerCo (Careil et al., 2024), we use the pretrained models implemented in the publicly available repositories [4]. For GLC (Jia et al., 2024), since no implementation is available, we report the results as presented in their paper.

## B    REVERSE-CHANNEL CODING

| **Algorithm 1 Sending** $\mathbf{x}_0$ (Ho et al., 2020) | **Algorithm 2 Receiving** |
|---|---|
| 1: Send $\mathbf{x}_T \sim q(\mathbf{x}_T\|\mathbf{x}_0)$ using $p(\mathbf{x}_T)$ | 1: Receive $\mathbf{x}_T$ using $p(\mathbf{x}_T)$ |
| 2: **for** $t = T - 1, \ldots, 2, 1$ **do** | 2: **for** $t = T - 1, \ldots, 1, 0$ **do** |
| 3:     Send $\mathbf{x}_t \sim q(\mathbf{x}_t\|\mathbf{x}_{t+1}, \mathbf{x}_0)$ using $p_\theta(\mathbf{x}_t\|\mathbf{x}_{t+1})$ | 3:     Receive $\mathbf{x}_t$ using $p_\theta(\mathbf{x}_t\|\mathbf{x}_{t+1})$ |
| 4: **end for** | 4: **end for** |
| 5: Send $\mathbf{x}_0$ using $p_\theta(\mathbf{x}_0\|\mathbf{x}_1)$ | 5: **return** $\mathbf{x}_0$ |

This section provides a brief overview of the reverse-channel coding (RCC) used in this paper. Algorithm 1 and Algorithm 2 illustrate the core procedure for transmitting a random sample $x \sim q(x)$ using a shared prior distribution $p(x)$. The objective is to communicate the sample $x$ using approximately $D_{KL}(q\|\|p)$ bits, leveraging a shared source of randomness between the sender and receiver. This problem, referred to as RCC in information theory, is addressed in our framework using the Poisson Functional Representation (PFR) algorithm (Theis & Ahmed, 2022), as detailed in Algorithm 3. PFR enables exact sampling from the target distribution while requiring only marginally more than $D_{KL}(q\|\|p)$ bits. For a more detailed theoretical analysis and proof of this algorithm, please refer to Theis & Ahmed (2022); Vonderfecht & Liu (2025).

---

**Algorithm 3 PFR** Encoding (Theis & Ahmed, 2022)

**Require:** $p, q, w_{\min}$
1: $t, n, s^* \leftarrow 0, 1, \infty$
2: **repeat**
3:     $z \leftarrow \mathtt{simulate}(n, p)$                          ▷ Candidate generation
4:     $t \leftarrow t + \mathtt{expon}(n, 1)$                        ▷ Poisson process
5:     $s \leftarrow t \cdot p(z)/q(z)$                            ▷ Candidate's score
6:     **if** $s \leq s^*$ **then**                        ▷ Accept/reject candidate
7:         $s^*, n^* \leftarrow s, n$
8:     **end if**
9:     $n \leftarrow n + 1$
10: **until** $s^* \leq t \cdot w_{\min}$
11: **return** $n^*$

---

**Algorithm 4 PFR** Decoding (Theis & Ahmed, 2022)

**Require:** $n^*, p$
1: **return** $\mathtt{simulate}(n^*, p)$

---

In Algorithm 3 and Algorithm 4, simulate denotes a shared pseudorandom generator that, given a random seed $n$ and a distribution $p$, produces a pseudorandom sample $z \sim p$.

---

[1] https://github.com/facebookresearch/NeuralCompression/tree/main/projects/illm
[2] https://github.com/huai-chang/DiffEIC
[3] https://github.com/JeremyIV/diffc
[4] https://github.com/Nikolai10/PerCo

## C   MORE QUANTITATIVE RESULTS

In Figure 8∼10 and Table 1, we present comprehensive comparisons on the Kodak, DIV2K and CLIC2020 datasets using PSNR, SSIM, MS-SSIM, DISTS (Ding et al., 2022), LPIPS (Zhang et al., 2018), CLIPSim (Radford et al., 2021), FID (Heusel et al., 2017), MUSIQ (Ke et al., 2021), and CLIP-IQA (Wang et al., 2023). We compare against a diverse set of representative approaches, including GAN-based MS-ILLM (Muckley et al., 2023), tokenizer-based GLC (Jia et al., 2024) and DLF (Xue et al., 2025), as well as recent diffusion-based codecs such as DiffEIC (Li et al., 2025c), PerCo (Careil et al., 2024), OSCAR (Guo et al., 2025), StableCodec (Zhang et al., 2025a), ResULIC (Ke et al., 2025), RDEIC (Li et al., 2025d), and DiffC (Vonderfecht & Liu, 2025). Our method achieves substantial improvements across structure-oriented metrics (SSIM, MS-SSIM) and semantic fidelity metrics (LPIPS, DISTS, CLIPSim), especially in the ultra-low bitrates. Moreover, we observe significant gains in FID, which measures the distributional difference between reconstructed and original image sets, indicating enhanced consistency in our reconstructions. While our method also performs strongly on no-reference metrics such as MUSIQ and CLIP-IQA, it may not always achieve the best scores. This is likely because these metrics focus primarily on perceptual attributes like image sharpness, without explicitly considering fidelity to the original content. Importantly, unlike existing diffusion- and GAN-based codecs that require dedicated perceptual training or multi-stage finetuning, our perceptual reconstruction is entirely training-free, offering both stronger performance and significantly reduced system complexity.

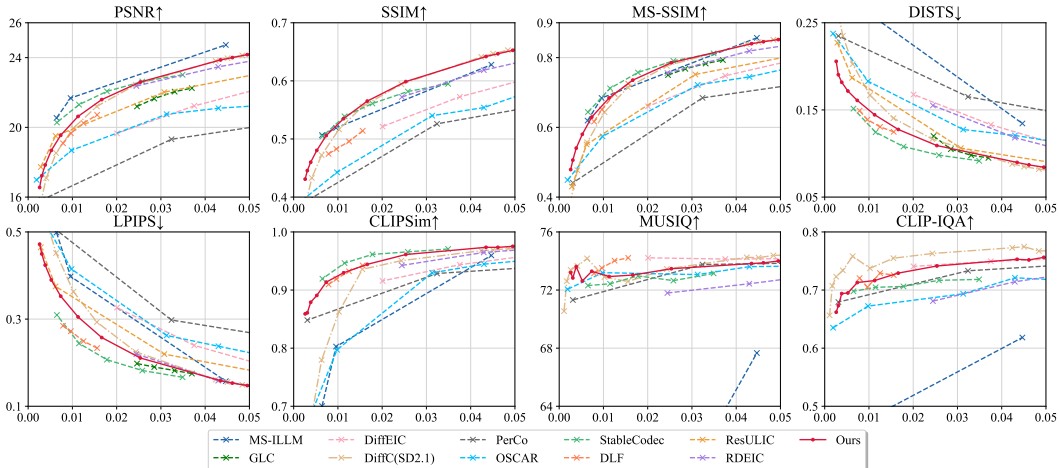

Figure 8: Rate-metric comparisons with SOTA method on Kodak dataset.

## D   MORE QUALITATIVE RESULTS

Figure 12 presents qualitative comparisons on the Kodak dataset under both ultra-low and low bitrate settings. As shown, our method achieves significantly better frame-wise fidelity across both settings. In contrast, PerCo (Careil et al., 2024) and DiffC (Vonderfecht & Liu, 2025) struggle to preserve correct semantic content, even when operating at relatively higher bitrates within the ultra-low range. Under low bitrate conditions, although both DiffEIC (Li et al., 2025c) and PerCo are able to reconstruct semantically plausible images, their results lack the visual consistency and detail fidelity achieved by our approach.

## E   QUALITATIVE RESULTS AT DIFFERENT BITRATES

By controlling the number of diffusion trajectories transmitted in the RCC algorithm, the compressed bitrate can be naturally adjusted. Figure 13 below presents visual results of our method at different bitrates. In the examples, the explicit component is fixed at 0.00206 bpp, while the implicit component is progressively encoded. At the lowest bitrate, the reconstructed image preserves the overall semantics but exhibits noticeable differences in fine details compared to the original.

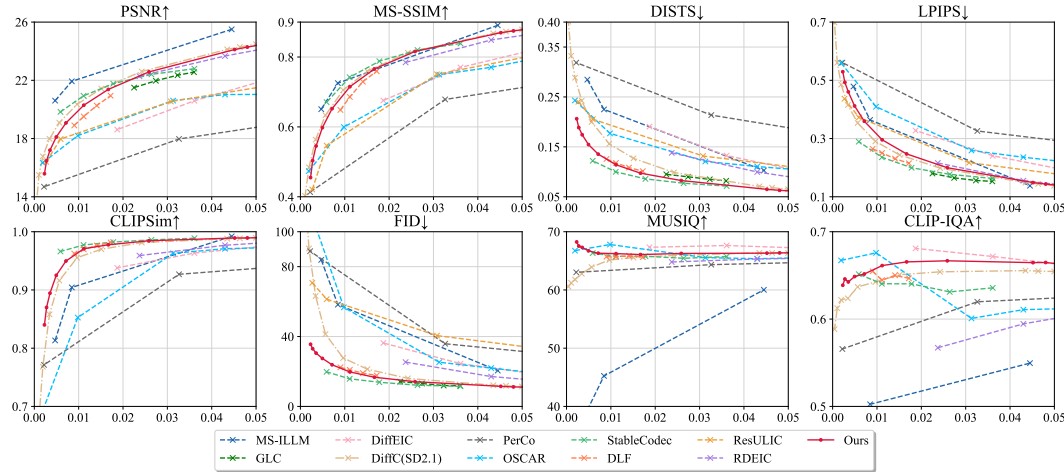

Figure 9: Rate-metric comparisons with SOTA method on DIV2K dataset.

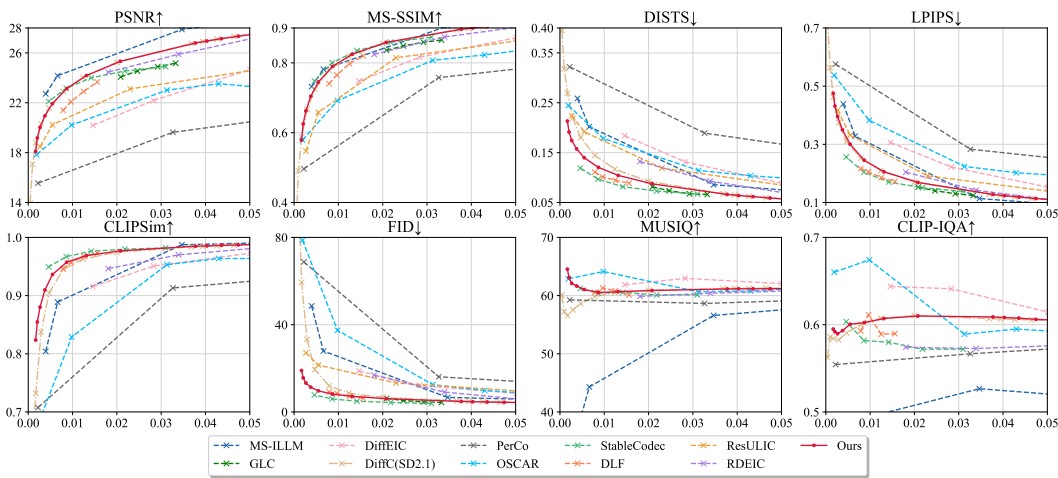

Figure 10: Rate-metric comparisons with SOTA method on CLIC2020 dataset.

As the bitrate of the implicit component increases, these differences gradually diminish, and the reconstruction becomes nearly indistinguishable from the ground truth.

## F    RATE ALLOCATION

Figure 14 below shows visual results at the same total bitrate but with different proportions of explicit and implicit components. It can be observed that allocating only a small portion of the bitrate to the explicit component yields superior visual quality, which is consistent with the findings in Figure 5 and Figure 15. This indicates that a small amount of explicit information effectively complements the implicit representation. Based on this observation, we adopt the lowest-bitrate point of Perco as our baseline model.

## G    RATE-DISTORTION-PERCEPTION TRADEOFF

Figure 17 presents visual results across different bitrates and tradeoff levels $\tau$. As shown in the low-bitrate examples in Figure 17a, setting $\tau = 0$ prioritizes PSNR by suppressing high-frequency details, resulting in overly smooth reconstructions that favor pixel-level fidelity. In contrast, setting

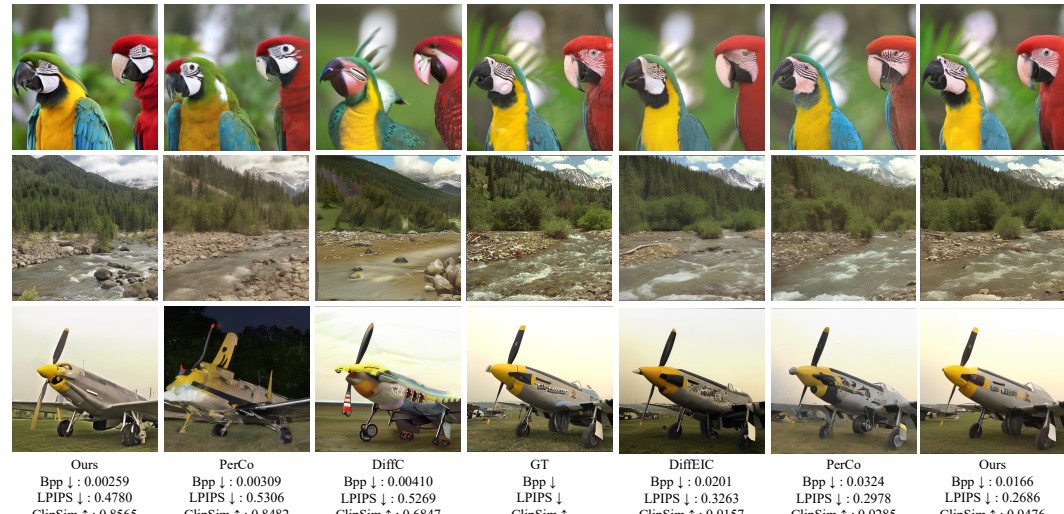

| Ours | PerCo | DiffC | GT | DiffEIC | PerCo | Ours |
|---|---|---|---|---|---|---|
| Bpp ↓ : 0.00259 | Bpp ↓ : 0.00309 | Bpp ↓ : 0.00410 | Bpp ↓ | Bpp ↓ : 0.0201 | Bpp ↓ : 0.0324 | Bpp ↓ : 0.0166 |
| LPIPS ↓ : 0.4780 | LPIPS ↓ : 0.5306 | LPIPS ↓ : 0.5269 | LPIPS ↓ | LPIPS ↓ : 0.3263 | LPIPS ↓ : 0.2978 | LPIPS ↓ : 0.2686 |
| ClipSim ↑ : 0.8565 | ClipSim ↑ : 0.8482 | ClipSim ↑ : 0.6847 | ClipSim ↑ | ClipSim ↑ : 0.9157 | ClipSim ↑ : 0.9285 | ClipSim ↑ : 0.9476 |

Figure 11: Qualitative comparisons of different methods on Kodak dataset. Best viewed on screen for details.

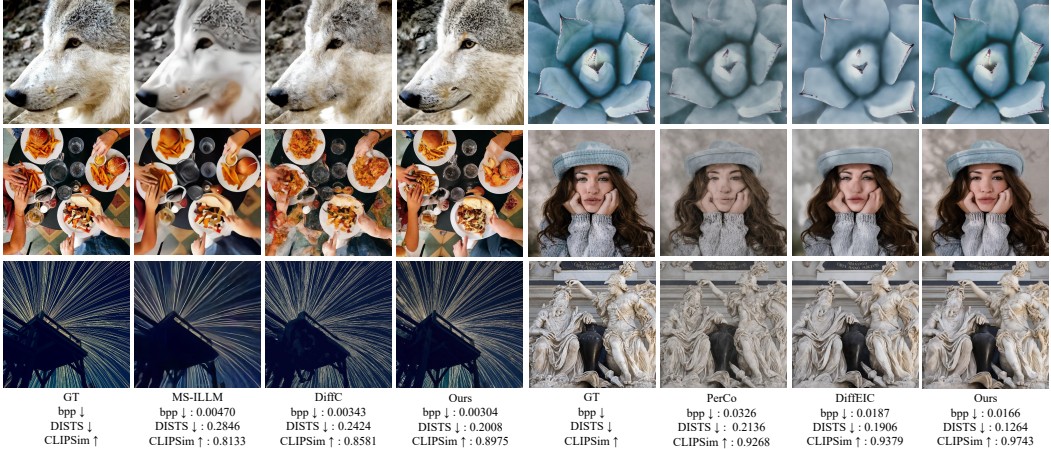

| GT | MS-ILLM | DiffC | Ours | GT | PerCo | DiffEIC | Ours |
|---|---|---|---|---|---|---|---|
| bpp ↓ | bpp ↓ : 0.00470 | bpp ↓ : 0.00343 | bpp ↓ : 0.00304 | bpp ↓ | bpp ↓ : 0.0326 | bpp ↓ : 0.0187 | bpp ↓ : 0.0166 |
| DISTS ↓ | DISTS ↓ : 0.2846 | DISTS ↓ : 0.2424 | DISTS ↓ : 0.2008 | DISTS ↓ | DISTS ↓ : 0.2136 | DISTS ↓ : 0.1906 | DISTS ↓ : 0.1264 |
| CLIPSim ↑ | CLIPSim ↑ : 0.8133 | CLIPSim ↑ : 0.8581 | CLIPSim ↑ : 0.8975 | CLIPSim ↑ | CLIPSim ↑ : 0.9268 | CLIPSim ↑ : 0.9379 | CLIPSim ↑ : 0.9743 |

Figure 12: Qualitative comparisons of different methods on DIV2K dataset. Best viewed on screen for details.

$\tau = 1$ yields lower PSNR scores but better preserves high-frequency content, leading to reconstructions that align more closely with human perceptual preferences and frame-wise fidelity. A similar trend is observed in the ultra-low bitrate as shown in Figure 17b. While $\tau = 0$ typically leads to overly smooth and blurry reconstructions due to the suppression of high-frequency details, this effect is less pronounced at ultra-low bitrates. Even with $\tau = 0$, the model still produces visually satisfactory results with acceptable sharpness. In comparison, $\tau = 1$ further enhances perceptual quality by restoring more high-frequency details. Furthermore, varying $\tau$ has only a marginal effect on the bitrate, which remains within a similar fluctuation range across different tradeoff settings.

To further quantify the effect of the tradeoff parameter on the distortion–perception balance, we conduct controlled experiments on the DIV2K dataset and report results using two complementary perceptual metrics, LPIPS and FID, as shown in Figure 16. The curves clearly illustrate how varying $\tau$ shifts the operating point between pixel-level fidelity and perceptual quality, enabling smooth and continuous control under similar bitrate ranges.

## H  PARTITIONING STRATEGY

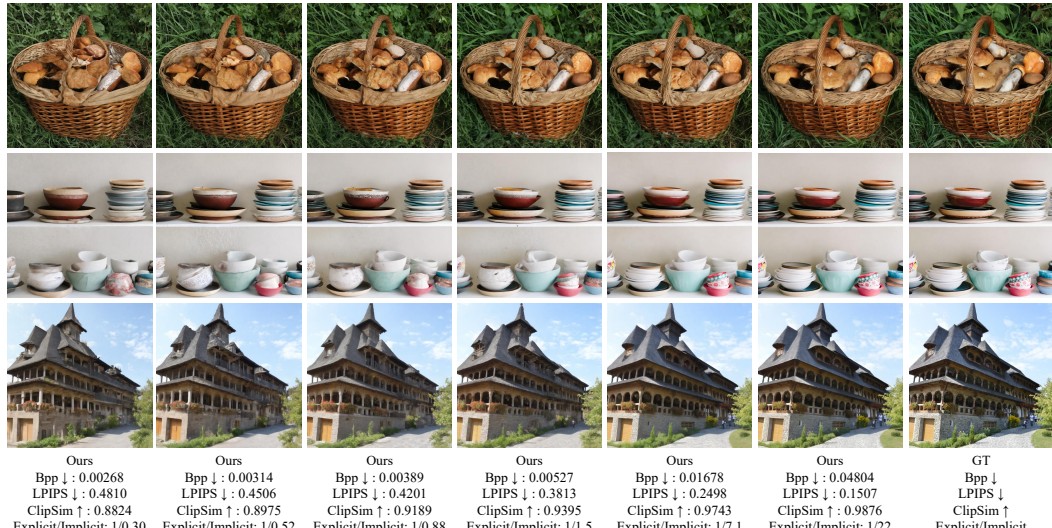

| Ours | Ours | Ours | Ours | Ours | Ours | GT |
|---|---|---|---|---|---|---|
| Bpp ↓ : 0.00268 | Bpp ↓ : 0.00314 | Bpp ↓ : 0.00389 | Bpp ↓ : 0.00527 | Bpp ↓ : 0.01678 | Bpp ↓ : 0.04804 | Bpp ↓ |
| LPIPS ↓ : 0.4810 | LPIPS ↓ : 0.4506 | LPIPS ↓ : 0.4201 | LPIPS ↓ : 0.3813 | LPIPS ↓ : 0.2498 | LPIPS ↓ : 0.1507 | LPIPS ↓ |
| ClipSim ↑ : 0.8824 | ClipSim ↑ : 0.8975 | ClipSim ↑ : 0.9189 | ClipSim ↑ : 0.9395 | ClipSim ↑ : 0.9743 | ClipSim ↑ : 0.9876 | ClipSim ↑ |
| Explicit/Implicit: 1/0.30 | Explicit/Implicit: 1/0.52 | Explicit/Implicit: 1/0.88 | Explicit/Implicit: 1/1.5 | Explicit/Implicit: 1/7.1 | Explicit/Implicit: 1/22 | Explicit/Implicit |

Figure 13: Qualitative comparisons of different bitrates on DIV2K dataset. Best viewed on screen for details.

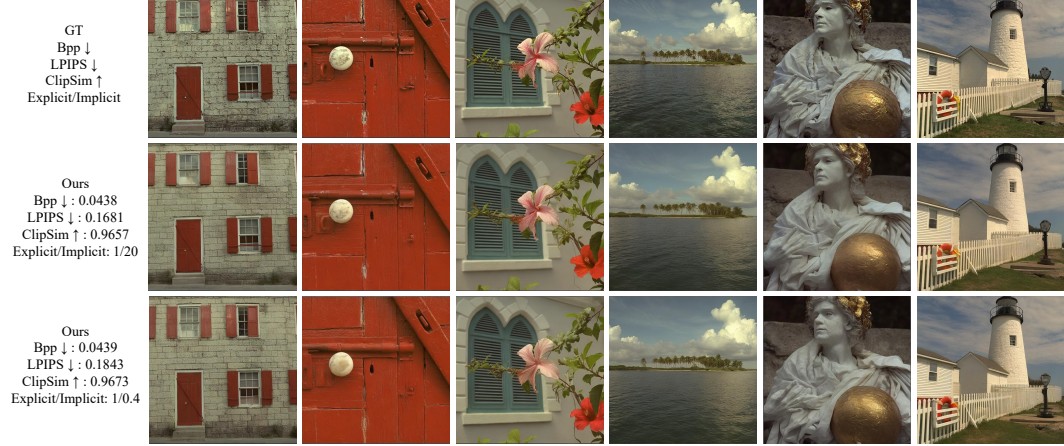

Figure 14: Qualitative comparison of different bitrate allocation strategies on the Kodak dataset. Best viewed on screen for details.

We evaluate different tile sizes in Table 5. All tiling strategies substantially improve performance compared with no tiling. A tile size of 512 pixels provides the best perceptual fidelity, achieving the largest BD-Rate reductions in DISTS and FID. In contrast, a tile size of 768 pixels yields the best semantic fidelity, achieving the lowest CLIPSim BD-Rate, though its DISTS and FID improvements are

Table 5: BD-Rate (%) under different tile sizes on DIV2K.

| | w/o | 512 | 768 | 1024 |
|---|---|---|---|---|
| **DISTS** | 0.00 | **-66.63** | -44.20 | -39.98 |
| **CLIPSim** | 0.00 | -31.63 | **-38.28** | -19.94 |
| **FID** | 0.00 | **-65.81** | -50.19 | -46.53 |

slightly weaker than the 512-pixel setting. These results indicate that different tile sizes offer distinct strengths, and that tile-based processing not only reduces memory consumption but can also enhance reconstruction quality by operating closer to the model's trained resolution.

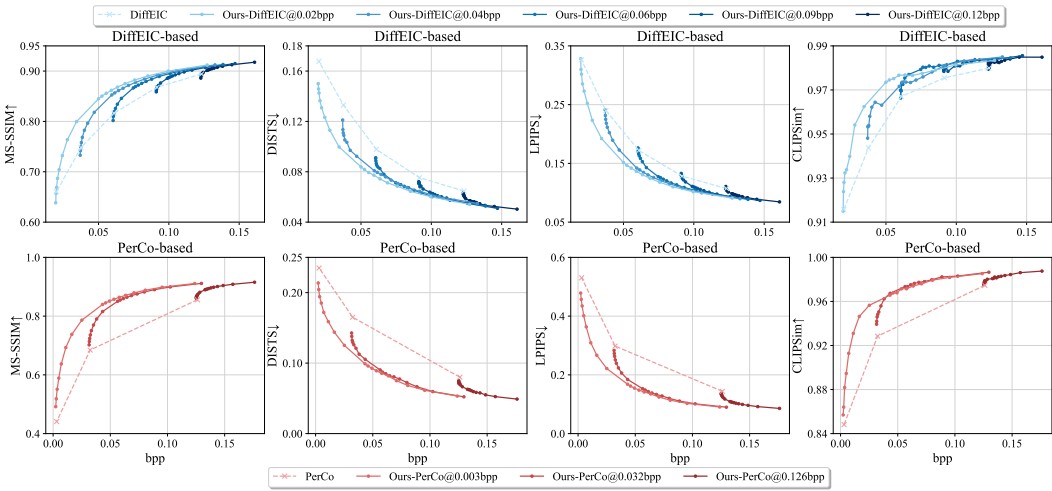

Figure 15: Rate–metric curves on the Kodak dataset. Our method is applied to multiple versions of each base model, trained at different bitrates, resulting in several curves per model. Our method consistently outperforms the corresponding baselines across all bitrate settings.

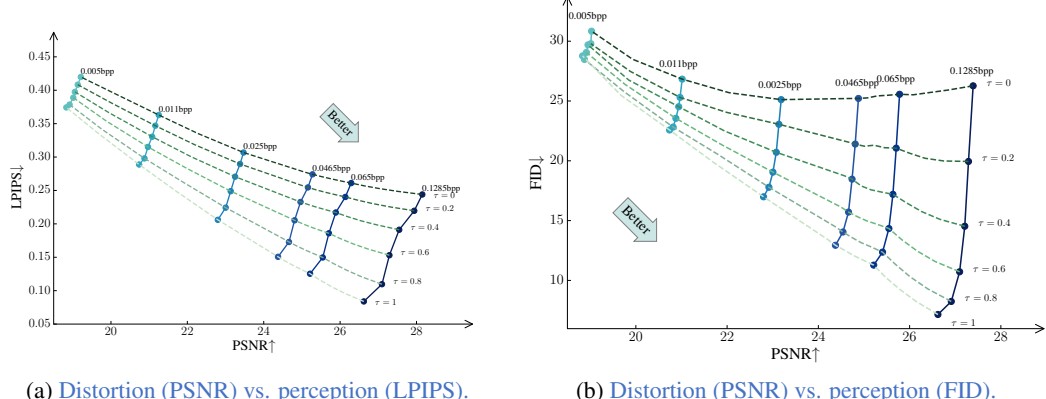

(a) Distortion (PSNR) vs. perception (LPIPS).  (b) Distortion (PSNR) vs. perception (FID).

Figure 16: Rate–distortion-perception tradeoff curves on the DIV2K dataset. Dashed lines indicate results with the same $\tau$, while solid lines connect results with similar bitrates.

# I   THE USE OF LARGE LANGUAGE MODELS (LLMS)

This paper uses a large language model solely to assist with minor language polishing and grammar refinement. All research ideas, experimental designs, analyses, and conclusions are developed entirely by the authors.

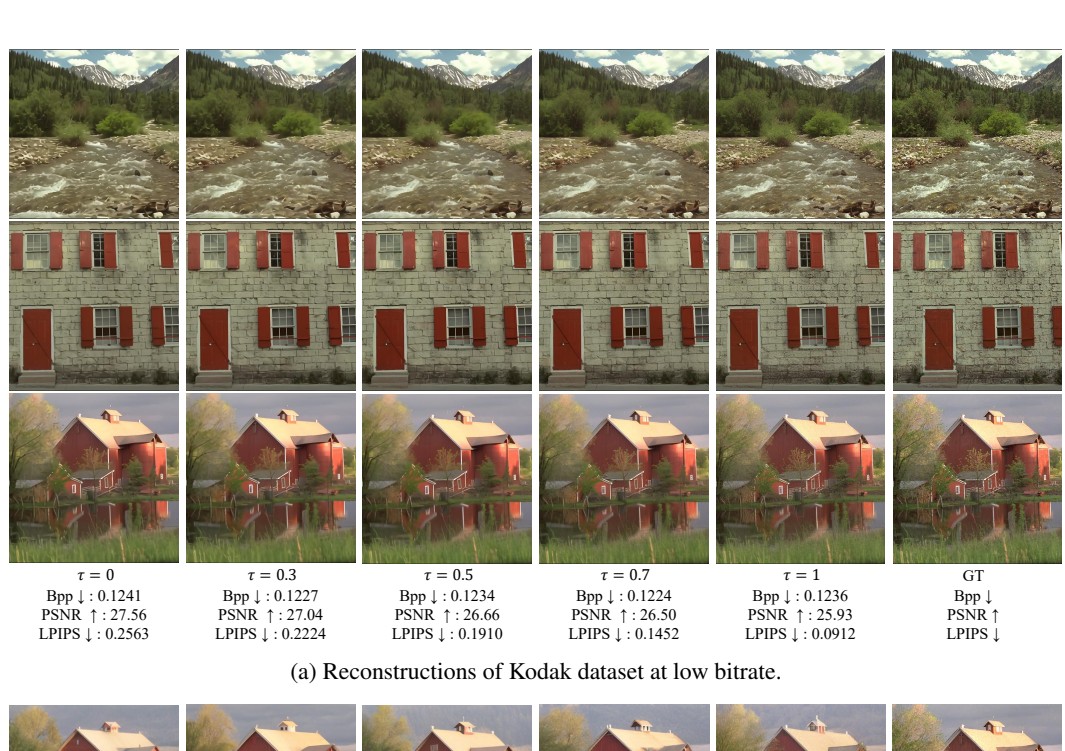

| $\tau = 0$ | $\tau = 0.3$ | $\tau = 0.5$ | $\tau = 0.7$ | $\tau = 1$ | GT |
|---|---|---|---|---|---|
| Bpp ↓ : 0.1241 | Bpp ↓ : 0.1227 | Bpp ↓ : 0.1234 | Bpp ↓ : 0.1224 | Bpp ↓ : 0.1236 | Bpp ↓ |
| PSNR ↑ : 27.56 | PSNR ↑ : 27.04 | PSNR ↑ : 26.66 | PSNR ↑ : 26.50 | PSNR ↑ : 25.93 | PSNR ↑ |
| LPIPS ↓ : 0.2563 | LPIPS ↓ : 0.2224 | LPIPS ↓ : 0.1910 | LPIPS ↓ : 0.1452 | LPIPS ↓ : 0.0912 | LPIPS ↓ |

(a) Reconstructions of Kodak dataset at low bitrate.

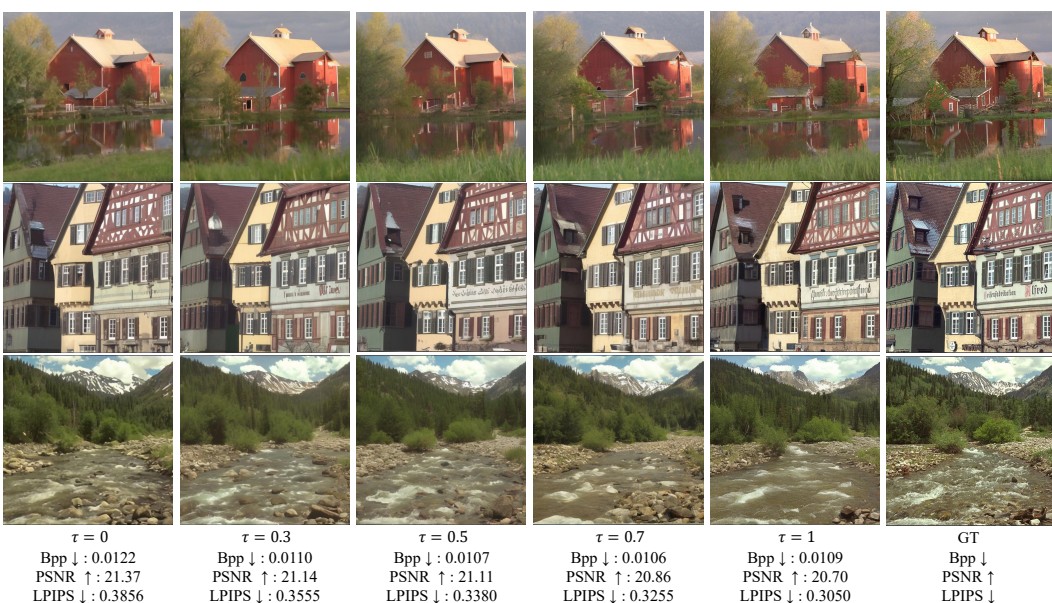

| $\tau = 0$ | $\tau = 0.3$ | $\tau = 0.5$ | $\tau = 0.7$ | $\tau = 1$ | GT |
|---|---|---|---|---|---|
| Bpp ↓ : 0.0122 | Bpp ↓ : 0.0110 | Bpp ↓ : 0.0107 | Bpp ↓ : 0.0106 | Bpp ↓ : 0.0109 | Bpp ↓ |
| PSNR ↑ : 21.37 | PSNR ↑ : 21.14 | PSNR ↑ : 21.11 | PSNR ↑ : 20.86 | PSNR ↑ : 20.70 | PSNR ↑ |
| LPIPS ↓ : 0.3856 | LPIPS ↓ : 0.3555 | LPIPS ↓ : 0.3380 | LPIPS ↓ : 0.3255 | LPIPS ↓ : 0.3050 | LPIPS ↓ |

(b) Reconstructions of Kodak dataset at ultra-low bitrate.

Figure 17: Comparison of reconstructions across different distortion–perception tradeoff levels ($\tau$)

