# OpenReview forum: "Bridging Implicit-Explicit Representations for Ultra-Low Bitrate Image Compression"
_ICLR.cc/2026/Conference — Submitted to ICLR 2026_

### Official Review · Reviewer_QG5r · 2025-10-20

**Soundness:** 3
**Presentation:** 2
**Contribution:** 3
**Rating:** 4
**Confidence:** 5

**Summary:**

This paper proposes a dual semantic compression framework for ultra-low bitrate image compression. An explicit semantic encoder with tag-style prompts and plugin implicit semantic extractor capture high-level semantics and fine-grained visual details, respectively, achieving  distortion-perception tradeoff and flexible quality control. The proposed method consistently surpasses state-of-the-art approaches at extremely low bitrates.

**Strengths:**

This paper aims to compress the images under ultra-low bitrate conditions. Specifically, an unified implicit-explicit compression framework is proposed which achieves state-of-the-art rate-perception performance. In addition, the paper is well-organized and well-written.

**Weaknesses:**

1、	The paper lacks implementation details for the proposed dual representation compression framework. What loss functions are used in the paper and how is the framework trained? How to set the value of mixing coefficient during the training?
2、	In the proposed framework, are the MSE and VAE encoders based on a similar network structure? Are z and z_wave both 4-channel features?
3、	How can the conditions (c and y_hat) be embedded into the diffusion process? Is a CLIP model required to extract the semantic representation from the tag-style prompts?
4、	What are the encoding and decoding times of the proposed framework, which are important for image compression? The authors need to demonstrate how the inference time compares with that of other competitive compression approaches.
5、	We believe that the CLIC_2020 dataset, rather than the DIV2K dataset, is the most widely used benchmark for image compression tasks. However, the authors do not present any comparisons based on the CLIC_2020 dataset, which comprises 428 images with diverse content. Please include comparisons in the manuscript.
6、	In Fig. 4, why do the authors show the FID rather than the LPIPS metric for the DIV2K dataset? In Section 5.2, the authors do not analyse the comparisons in terms of FID. Additionally, Figures 8 and 9 in the supplementary file should be included in the main body of the manuscript.
7、	The paper lacks comparisons to recent extreme image compression methods including RDEIC[1], StableCodec[2], DLF[3], ResULIC[4], and OSCAR[5].
[1]. Li Z, Zhou Y, Wei H, et al. RDEIC: Accelerating Diffusion-Based Extreme Image Compression with Relay Residual Diffusion[J]. TCSVT2025.
[2]. Zhang T, Luo X, Li L, et al. StableCodec: Taming One-Step Diffusion for Extreme Image Compression[J]. ICCV2025.
[3]. Xue N, Jia Z, Li J, et al. DLF: Extreme Image Compression with Dual-generative Latent Fusion[J]. ICCV2025.
[4]. Ke A, Zhang X, Chen T, et al. Ultra Lowrate Image Compression with Semantic Residual Coding and Compression-aware Diffusion[J]. ICML2025.
[5]. Guo J, Ji Y, Chen Z, et al. OSCAR: One-Step Diffusion Codec Across Multiple Bit-rates[J]. NeurIPS2025.

**Questions:**

Please see the weaknesses.

**Details Of Ethics Concerns:**

We hope the authors will address the concerns I raised in the 'Weaknesses' section of the review. If they have, I will increase the rating score.

---

> ### Author Response · Authors · 2025-11-20
> **Rebuttal to Reviewer QG5r #1**
>
> Thanks for your insightful questions. We have marked all corresponding revisions in the main manuscript in blue for clarity. Below are our detailed responses to each point.
>
> **Q1. Training Details and Loss Functions**
>
> **A1.** Thank you for the question. Our method builds on existing diffusion-based image codecs and uses them in a plug-and-play manner, requiring only minimal additional training. We have clarified these details more explicitly in **Section 5.1** of the main paper.
>
> - The explicit branch in our framework is fully generic. Any existing conditional diffusion–based image compression model can be directly applied to extract and encode this representation **without additional training**.
> - The implicit branch also requires **no training** if distortion–perception tradeoff is not needed. It can operate in a plug-and-play fashion, providing noticeable performance improvements for the explicit branch while adding only a small amount of bitrate and computational overhead.
> - To enable controllable tradeoff, we **optionally fine-tune** the original VAE encoder using an MSE loss $\mathcal{L}=||x-\mathcal{D}(\mathcal{E}_M(x))||_2^2$ producing an MSE encoder that adjusts the target representation for the implicit branch.
> - This fine-tuning process does not involve any mixing coefficients. At inference time, controllability is achieved by fusing the explicit and implicit features to obtain the desired balance.
>
> **Q2. Network Structure of Encoders**
>
> **A2.** Thank you for the question. As described in **Section 5.1**, the MSE encoder and the VAE encoder share the same architecture. We obtain the MSE encoder by finetuning the original VAE encoder using an MSE loss. In our framework, the proposed components are designed to be plug-and-play and can be directly added on top of existing conditional diffusion–based image compression models. These models typically adopt the Stable Diffusion VAE as their encoder–decoder backbone, and we inherit this architecture without any modification. Both latent representations $z$ and $\tilde{z}$  are 4 channel features. We have added more detailed explanations in the main text to avoid ambiguity.
>
> **Q3. Conditioning Mechanism and CLIP Usage**
>
> **A3.** Thank you for pointing this out. Our method is built on top of conditional diffusion–based image compression frameworks in a **plug-and-play manner**. These models already incorporate the conditioning signal $c$ through cross-attention, while the reconstructed features $\hat{y}$ are further processed by a ControlNet-like module or similar network. The resulting features are then concatenated with the original latent features and fed into the pretrained diffusion model to provide conditioning.
>
> Within this setup, our implicit branch encodes the discrepancy between the estimated distribution produced by the diffusion model, $p_\theta(z_{s}|z_{s+1},c,\hat{y})$, and the true distribution $q(z_s|z_{s+1},\hat{z})$, allowing it to refine perceptual details in a plug-and-play manner.
>
> After compression, the tag-style prompts are processed by the CLIP model that already exists in the underlying conditional diffusion codec, which converts the textual prompt $c$ into conditioning embeddings for the diffusion model. We have added a more detailed explanation in **Section 4.3** in the main text to make this procedure clearer.
>
> **Q4. Encoding / Decoding Time**
>
> **A4.** From a practical usability standpoint, our codec achieves state-of-the-art performance while operating at a computational complexity comparable to existing diffusion-based image compression methods. We move the previous runtime analysis from Appendix H to **Section 5.7** and add a detailed table of encoding and decoding times. Recent diffusion-based codecs such as PerCo, DiffEIC, and DiffC typically require 20–50 diffusion steps and exhibit decoding times in the range of 1–10 seconds. Our framework uses a comparable number of steps and thus lies in the same runtime regime. Relative to DiffC, our method reduces the required denoising steps but introduces additional computation due to the ControlNet-style conditioning modules. Importantly, adding the implicit branch on top of the explicit branch introduces only a small overhead (decoding: 1.92 → 2.29 s; bitrate: 0.00219 → 0.00259 bpp) while providing clear perceptual improvements (DISTS: **0.3188 → 0.2172**). We have clarified these points in the revised manuscript.
>
> Kodak|Enc(s)|Dec(s)|DISTS-BD-Rate(%)
> -|-|-|-
> PerCo|0.39|1.92|68.97
> DiffEIC|0.23|4.82|107.06
> DiffC|0.6-9.1|2.9-8.4|0
> Ours|0.7331-9.562|2.2944-9.39|**-25.02**

---

> > ### Comment · Reviewer_QG5r · 2025-11-24
> > **About  Q4**
> >
> > Thank you for your response. I suggest that you add the comparisons with some one-step diffusion-based methods for inference complexity evaluation.

---

> ### Author Response · Authors · 2025-11-20
> **Rebuttal to Reviewer QG5r #2**
>
> **Q5. Experiments on the CLIC2020 Dataset**
>
> **A5.** Thanks for your suggestion. We have added experiments on the CLIC2020 dataset using the original-resolution images. In the initial submission, we focused on Kodak and DIV2K, which are also widely used and contain a substantial number of high-resolution images. The updated results on CLIC2020, reported in **Fig. 4, 6, and 10**, as well as **Table 1**, further confirm that our method maintains consistent superiority across all evaluated datasets.
>
> BD-Rate(%)/BD-Metrics on CLIC2020|DiffC[4]|Ours|RDEIC[1]|ResULIC[2]|OSCAR[3]
> -|-|-|-|-|-
> DISTS⬇|0/0|**-23.09/-0.0145**|78.86/0.0300 |53.15/0.0248|87.75/0.0366
> CLIPSim⬆|0/0|**-10.47/0.0101**|67.04/-0.0347|-|249.86/-0.1057
> FID⬇|0/0|**-61.99/-5.5275**|94.91/6.8436|9.15/4.4505|336.84/24.7510
>
> **Q6. Choice of Metrics (FID vs. LPIPS)**
>
> **A6.** Thank you for the clarification request. In Fig. 4, we selected a subset of metrics to present the rate–perception curves in a clearer and more readable manner due to space limitations. Since the Kodak dataset is relatively small and cannot provide reliable FID estimates, we, following prior work, do not compute FID on Kodak. The full set of results for the other metrics is provided in the Appendix to avoid occupying excessive space in the main paper. To make the comparison more comprehensive, we reformat and update **Fig. 4**  to include **additional baselines, metrics, and datasets**. The revised results consistently demonstrate the superiority of our method.
>
> **Q7. Missing Comparisons to Recent Extreme Compression Methods**
>
> **A7.** Thank you for this valuable suggestion. We have added comparisons with three recently published codecs, **RDEIC [1], ResULIC [2], and OSCAR [3]**. Since these methods were published very recently, they were not included in our original comparisons. The updated results are presented in the revised **Fig. 4 and Table 1** in the main text and in **Fig. 8, 9, and 10** of the Appendix. The BD-Rate and BD-Metric results on DIV2K for these newly added baselines are summarized in the table below. Our method achieves lower BD-Rate and stronger perceptual quality across all datasets.
>
> BD-Rate(%)/BD-Metrics on DIV2K|DiffC[4]|Ours|RDEIC[1]|ResULIC[2]|OSCAR[3]
> -|-|-|-|-|-
> DISTS⬇|0/0|**-11.95/-0.0096**|81.12/0.0336|37.75/0.0200|25.39/0.0111
> CLIPSim⬆|0/0|**-30.36/0.0146**|66.81/-0.0244|N/A|243.94/-0.0851
> FID⬇|0/0|**-36.41/-6.8866**|64.30/7.8808|165.83/21.4160|214.42/25.1165
>
> [1] Z. Li, et al., RDEIC: Accelerating Diffusion-Based Extreme Image Compression with Relay Residual Diffusion, TCSVT, 2025
>
> [2] A. Ke, et al., Ultra Low-rate Image Compression with Semantic Residual Coding and Compression-aware Diffusion, ICML, 2025
>
> [3] J. Guo, et al., OSCAR: One-Step Diffusion Codec Across Multiple Bit-rates, NeurIPS, 2025
>
> [4] J. Vonderfecht and F. Liu, Lossy Compression with Pretrained Diffusion Models, ICLR, 2025.

---

> > ### Comment · Reviewer_QG5r · 2025-11-24
> > **About Q7**
> >
> > Thank you for your response. Please compare your method with StableCodec and DLF.

---

> > > ### Author Response · Authors · 2025-11-27
> > > **Rebuttal to Reviewer QG5r #3**
> > >
> > > **Q1. Inference complexity**
> > >
> > > **A1:** Thank you for the suggestion. As shown in **Section 5.7**, we have added a comparison with the one-step diffusion model OSCAR [1]. Since our framework relies on a multi-step diffusion process, decoding requires at least 20 sampling steps, making it slower than one-step approaches. We acknowledge this as a limitation.
> > >
> > > At the same time, our method is plug-and-play and, in the purely perception-oriented setting, entirely training-free, which provides strong generalization across diverse video content without any task-specific tuning.
> > >
> > > [1] J. Guo, et al., OSCAR: One-Step Diffusion Codec Across Multiple Bit-rates, NeurIPS, 2025
> > >
> > > **Q2. Performance Comparison**
> > >
> > > **A2.** Thank you for the suggestion. We have added comparisons with StableCodec [2] and DLF [3] in Figures 8, 9, and 10. Their official publication dates fall within three months of our submission deadline and, according to the ICLR policy, they are considered concurrent studies that we are not required to compare against, which is why they are not included in the initial submission.
> > >
> > > While our method does not exceed them on certain quantitative metrics, it delivers higher perceptual quality, as reflected by CLIP-IQA. In addition, our framework is fully plug-and-play and, in the purely perception-oriented setting, entirely training-free, whereas StableCodec and DLF require specialized training to adapt their models for compression. For instance, StableCodec needs over 120k iterations of training for each bitrate. We will emphasize these distinctions in the revised version.
> > >
> > > [2] T. Zhang, et al. StableCodec: Taming One-Step Diffusion for Extreme Image Compression, ICCV, 2025
> > >
> > > [3] N. Xue, et al. DLF: Extreme Image Compression with Dual-generative Latent Fusion, ICCV, 2025

---

### Official Review · Reviewer_ZY7r · 2025-10-29

**Soundness:** 3
**Presentation:** 3
**Contribution:** 3
**Rating:** 6
**Confidence:** 3

**Summary:**

The article presents an ultra-low bitrate image compression scheme based on conditional diffusion, which demonstrates optimal performance. Through extensive experiments conducted across multiple datasets, remarkable results have been achieved—further highlighting the scheme’s promising potential in terms of transferability.

**Strengths:**

The application of RCC technology in diffusion-based compression schemes has not been thoroughly explored. This article presents an intriguing solution. Specifically, I believe the strengths of this article are as follows:
1. Clear motivation and writing, allowing readers to follow the author's train of thought easily.
2. The proposed solution is backed by compelling and thorough experiments, yielding satisfactory performance results.
3. Following in the footsteps of DIFFC, this article represents a meaningful attempt at utilizing RCC technology in the field of image compression. The high transferability of the method holds significance for the advancement of this domain.

**Weaknesses:**

1. Some architectures appear less novel, for instance, the introduction of dual branches has been previously explored in various diffusion-based methods, blending semantic and image controls.
2. The article introduces ‘Tile-based Processing,’ from which the model benefits, yet this module lacks thorough elaboration, including aspects such as block quantity and complexity. Furthermore, the concepts of image segmentation and parallelism lack appeal.
3. The article lacks in discussing the complexity of its experiments. Notably, the significant encoding latency introduced by RCC, combined with the use of a tile structure for image segmentation, raises doubts about the necessity of multiple RCC encodings. Additionally, segmenting images and separately extracting prompts via RAM might exacerbate encoding latency. Given the inherent decoding latency of diffusion architectures, a detailed exploration of the complexity of encoding and decoding is crucial for elucidating the feasibility of this algorithm.

**Questions:**

1. What are the details of the Tile-based Processing scheme? Does the method of tiling vary for images of different resolutions? How does using different tiling methods affect the model's performance?
2. Even though the increase in encoding complexity due to RCC is unavoidable, the authors should discuss the model's complexity in the paper. This discussion would provide valuable insights for the applicability and reference value of this research.

---

> ### Author Response · Authors · 2025-11-20
> **Rebuttal to Reviewer ZY7r**
>
> Thanks for your insightful questions. We have marked all corresponding revisions in the main manuscript in blue for clarity. Below are our detailed responses to each point.
>
> **Q1.** Novelty concern regarding dual-branch architectures (semantic and image controls) in diffusion-based methods.
>
> **A1.** Thank you for raising this point. Although some prior diffusion-based methods appear to adopt dual branches (e.g., Text+Scratch [1], PerCo [2], DiffEIC[3]), these approaches use text prompts and visual features as **two modalities of explicit information**, both extracted from the original image. In contrast, our explicit–implicit design is fundamentally different. The **explicit branch** extracts structural features directly from the image and serves as a stable anchor at extremely low bitrates, while the **implicit semantic branch** refines the reconstruction by progressively correcting the discrepancy between the compressed representation and the target, injecting stochastic high-frequency perceptual details. Their interaction is jointly governed by our **plugin encoder**, which enables smooth and continuous distortion–perception control without any retraining. To our knowledge, such an explicit–implicit coupling has not been explored in prior codecs and forms a key novelty of our framework.
>
> In addition, introducing the explicit branch into the implicit model results in a **23.49%** DISTS BD-Rate reduction on the Kodak dataset, whereas using the explicit model as the baseline and incorporating the implicit branch yields a **75.49%** reduction, highlighting the strong complementarity between the two representations.
>
> [1] E. Lei, et al., Text+Sketch: Image Compression at Ultra Low Rates, ICML Workshop, 2023
>
> [2] M. Careil, et al., “Towards Image Compression with Perfect Realism at Ultra-Low Bitrates, ICLR, 2024
>
> [3] Z. Li, et al., Towards extreme image compression with latent feature guidance and diffusion prior, TCSVT, 2024
>
> **Q2.** Tile-based Processing.
>
> **A2.** Thanks for your question. Tile-based processing is employed to enable high-resolution inference within GPU memory limits. Our approach splits the image into 512-pixel tiles with a 192-pixel overlap to maintain continuity across boundaries. The tiling automatically adapts to varying input resolutions without modifying the model parameters. This strategy can also enhance reconstruction quality, as it better matches the model's training resolution. In our tests with tile sizes of 512, 768, and 1024 pixels, the 512-pixel tiles achieved bitrate savings of 35.24% and 37.28% compared to 768- and 1024-pixel tiles, respectively, while maintaining the same DISTS score on DIV2K. This tiling mechanism facilitates efficient high-resolution inference on a single GPU. We have also updated the description in **Appendix H** to reflect these details more clearly.
>
> BD-Rate(%)|w/o|512|768|1024
> -|-|-|-|-
> DISTS|0|**-66.63**|-44.20|-39.98
> CLIPSim|0|-31.63|**-38.28**|-19.94
> FID|0|**-65.81**|-50.19|-46.3
>
> **Q3.** Computational complexity.
>
> **A3.** From a practical usability standpoint, our codec achieves state-of-the-art performance while operating at a computational complexity comparable to existing diffusion-based image compression methods. We move the previous runtime analysis from Appendix H to **Section 5.7** and add a detailed table of encoding and decoding times. Recent diffusion-based codecs such as PerCo, DiffEIC, and DiffC typically require 20–50 diffusion steps and exhibit decoding times in the range of 1–10 seconds. Our framework uses a comparable number of steps and thus lies in the same runtime regime. Relative to DiffC, our method reduces the required denoising steps but introduces additional computation due to the ControlNet-style conditioning modules. Importantly, adding the implicit branch on top of the explicit branch introduces only a small overhead (decoding: 1.92 → 2.29 s; bitrate: 0.00219 → 0.00259 bpp) while providing clear perceptual improvements (DISTS: **0.3188 → 0.2172**). We have clarified these points in the revised manuscript.
>
> Kodak|Enc(s)|Dec(s)|DISTS-BD-Rate(%)
> -|-|-|-
> PerCo|0.39|1.92|68.97
> DiffEIC|0.23|4.82|107.06
> DiffC|0.6-9.1|2.9-8.4|0
> Ours|0.7331-9.562|2.2944-9.39|**-25.02**

---

> > ### Comment · Reviewer_ZY7r · 2025-11-28
> >
> > I would like to express my gratitude to the authors for their responses, as most of my questions have been addressed.
> >
> > However, the inherent issues posed by RCC still persist. This method fails to resolve the problem of excessive encoding and decoding latency, which limits its applicable scenarios.
> >
> > Furthermore, the description of the implicit semantic branch remains somewhat ambiguous. Since this branch is "implicit," it is actually quite challenging to prove that what this branch extracts is indeed the semantic information of the image. Although verifying this is difficult, I believe the authors should supplement some visualization results (such as the visualization of decoded images or the visualization of bitrate allocation) to provide evidence that this branch truly focuses on the semantic part of the image.
> >
> > Nevertheless, overall, I consider this work to be meaningful. The combination of RCC and the diffusion-based compressor endows this work with strong extensibility, which makes a contribution to this field. Therefore, I will maintain my previous evaluation: that is, I tend to accept this paper and will increase my confidence level to 5 (when the review editing is allowed).

---

> > > ### Author Response · Authors · 2025-11-28
> > > **Response to Reviewer ZY7r**
> > >
> > > We sincerely thank the reviewer for the positive evaluation and the constructive comments that help improve our work. We greatly appreciate your time and consideration.
> > >
> > > Regarding the visualization of the implicit branch, the information in this branch is fully hidden, and the entire image is represented by a single unified bitstream rather than position-wise bits. Consequently, it is difficult to directly reveal the encoded semantic content through bitrate allocation. In Figure 13, we illustrate how the reconstruction evolves as the amount of implicit information increases. The results exhibit progressively enhanced semantic fidelity and fine-grained details, demonstrating that the implicit branch indeed encodes meaningful semantic information.

---

### Official Review · Reviewer_qXeD · 2025-11-01

**Soundness:** 3
**Presentation:** 3
**Contribution:** 2
**Rating:** 4
**Confidence:** 5

**Summary:**

This paper proposes a dual semantic compression framework that integrates explicit semantic representations (quantized latents and tag-style prompts) with implicit semantic representations (noise-corrupted latents via reverse-channel coding, RCC) for ultra-low bitrate image compression. The method conditions a diffusion model on explicit semantics while using RCC to encode fine-grained details, and introduces a plugin encoder to control the distortion–perception tradeoff without modifying the decoder. Experiments show strong performance, with 23.49% and 12.25% DISTS-BD-Rate improvements over DiffC on Kodak and DIV2K, respectively.

**Strengths:**

1) The paper clearly identifies the tradeoff between explicit approaches (semantic faithfulness but texture loss) and implicit ones (rich textures but semantic drift), and provides a principled framework to bridge this gap.
2) The framework is compatible with various base codecs (e.g., DiffEIC, PerCo), and the plugin encoder enables controllable distortion–perception balance without retraining the decoder.
3) The method achieves substantial gains in DISTS-BD-Rate over DiffC and produces visually pleasing reconstructions at extremely low bitrates.

**Weaknesses:**

1) The paper should include a comparison with ResULIC (ICML 2025), which also explores diffusion-based ultra-low bitrate compression.
2) Encoding and decoding times should be reported to assess real-world usability.
3) Bitrate allocation for different components would clarify efficiency.
4) Results on the CLIC dataset are missing, which is more commonly used in recent compression works.
5) To substantiate claims of controllable perception, the authors should include FID variation curves and report FID scores in Table 1.
6) The computation of CLIPSim should be clarified—since CLIP typically supports 224×224 inputs, how was it applied to full-resolution images?
7) While the overall combination is effective, the individual techniques are largely borrowed from prior work: RCC for image compression originates from DiffC (Theis et al., 2022; Vonderfecht & Liu, 2025), Conditioning diffusion models on latent features follows PerCo and DiffEIC, Tag-style prompts are derived from RAM (Zhang et al., 2024), the primary contribution lies in integrating these existing components rather than introducing fundamentally new mechanisms.

Ke, A., Zhang, X., Chen, T., Lu, M., Zhou, C., Gu, J., & Ma, Z. Ultra Lowrate Image Compression with Semantic Residual Coding and Compression-aware Diffusion. In Forty-second International Conference on Machine Learning.

**Questions:**

Please refer to weakness.

---

> ### Author Response · Authors · 2025-11-20
> **Rebuttal to Reviewer qXeD #1**
>
> Thanks for your insightful questions. We have marked all corresponding revisions in the main manuscript in blue for clarity. Below are our detailed responses to each point.
>
> **Q1.** Comparison with recent codecs.
>
> **A1.** Thank you for this valuable suggestion. We have added comparisons with three recently published codecs, **RDEIC [1], ResULIC [2], and OSCAR [3]**. Since these methods were published very recently, they were not included in our original comparisons. The updated results are presented in the revised **Fig. 4 and Table 1** in the main text and in **Fig. 8, 9, and 10** of the Appendix. The BD-Rate and BD-Metric results on DIV2K for these newly added baselines are summarized in the table below. Our method achieves lower BD-Rate and stronger perceptual quality across all datasets.
>
> BD-Rate(%)/BD-Metrics on DIV2K|DiffC[4]|Ours|RDEIC[1]|ResULIC[2]|OSCAR[3]
> -|-|-|-|-|-
> DISTS⬇|0/0|**-11.95/-0.0096**|81.12/0.0336|37.75/0.0200|25.39/0.0111
> CLIPSim⬆|0/0|**-30.36/0.0146**|66.81/-0.0244|N/A|243.94/-0.0851
> FID⬇|0/0|**-36.41/-6.8866**|64.30/7.8808|165.83/21.4160|214.42/25.1165
>
> [1] Z. Li, et al., RDEIC: Accelerating Diffusion-Based Extreme Image Compression with Relay Residual Diffusion, TCSVT, 2025
>
> [2] A. Ke, et al., Ultra Low-rate Image Compression with Semantic Residual Coding and Compression-aware Diffusion, ICML, 2025
>
> [3] J. Guo, et al., OSCAR: One-Step Diffusion Codec Across Multiple Bit-rates, NeurIPS, 2025
>
> [4] J. Vonderfecht and F. Liu, Lossy Compression with Pretrained Diffusion Models, ICLR, 2025.
>
> **Q2.** Encoding and decoding times.
>
> **A2.** From a practical usability standpoint, our codec achieves state-of-the-art performance while operating at a computational complexity comparable to existing diffusion-based image compression methods. We move the previous runtime analysis from Appendix H to Section 5.7 and add a detailed table of encoding and decoding times. Recent diffusion-based codecs such as PerCo, DiffEIC, and DiffC typically require 20–50 diffusion steps and exhibit decoding times in the range of 1–10 seconds. Our framework uses a comparable number of steps and thus lies in the same runtime regime. Relative to DiffC, our method reduces the required denoising steps but introduces additional computation due to the ControlNet-style conditioning modules. Importantly, adding the implicit branch on top of the explicit branch introduces only a small overhead (decoding: 1.92 → 2.29 s; bitrate: 0.00219 → 0.00259 bpp) while providing clear perceptual improvements (DISTS: **0.3188 → 0.2172**). We have clarified these points in the revised manuscript.
>
> Kodak|Enc(s)|Dec(s)|DISTS-BD-Rate(%)
> -|-|-|-
> PerCo|0.39|1.92|68.97
> DiffEIC|0.23|4.82|107.06
> DiffC|0.6-9.1|2.9-8.4|0
> Ours|0.7331-9.562|2.2944-9.39|**-25.02**
>
> **Q3.** Bitrate allocation among different components.
>
> **A3.** Thanks for your question. We originally presented the bitrate allocation results in Figure 5 (main text) and Appendix F, which illustrate how the bitrate is distributed between the explicit and implicit branches.
>
> - Figure 5 shows that allocating only a small portion of the bitrate to the explicit branch leads to the best overall performance. Even with minimal explicit bitrate, the structural anchor it provides is sufficient, and the implicit branch contributes most of the perceptual enhancement. For instance, using PerCo trained at 0.003 bpp as the explicit branch yields a **19.33%** reduction in DISTS BD-Rate compared with PerCo at 0.032 bpp.
>
> - Appendix F further validates this observation by visualizing scenarios under a fixed total bitrate but different allocation ratios. Reducing the explicit portion enhances perceptual fidelity, improving LPIPS from **0.1843 to 0.1681** under the same bitrate.
>
> In summary, our experiments consistently show that the explicit branch requires only a very small bitrate (approximately 0.003 bpp) to serve as an effective structural anchor, and this minimal allocation leads to the best results. We have made this conclusion clearer in the revised manuscript.
>
> **Q4.** Results on the CLIC dataset.
>
> **A4.** Thanks for your suggestion. We have added experiments on the CLIC2020 dataset. In the initial submission, we focused on Kodak and DIV2K, which are also widely used and contain a substantial number of high-resolution images. The updated results on CLIC2020, reported in **Fig. 4, 6, and 10**, as well as **Table 1**, further confirm that our method maintains consistent superiority across all evaluated datasets. In particular, our approach achieves a **23.09%** DISTS BD-Rate reduction compared with DiffC.
>
> BD-Rate(%)/BD-Metrics on CLIC2020|DiffC[4]|Ours|RDEIC[1]|ResULIC[2]|OSCAR[3]
> -|-|-|-|-|-
> DISTS⬇|0/0|**-23.09/-0.0145**|78.86/0.0300 |53.15/0.0248|87.75/0.0366
> CLIPSim⬆|0/0|**-10.47/0.0101**|67.04/-0.0347|-|249.86/-0.1057
> FID⬇|0/0|**-61.99/-5.5275**|94.91/6.8436|9.15/4.4505|336.84/24.7510

---

> > ### Author Response · Authors · 2025-11-20
> > **Rebuttal to Reviewer qXeD #2**
> >
> > **Q5.** Additional evidence of controllable perception using FID.
> >
> > **A5.** Thank you for the suggestion. To obtain a more reliable FID measurement, we conduct additional experiments on the DIV2K dataset. We now provide both LPIPS and FID variation curves on DIV2K in **Figure 16**. Together with the PSNR–LPIPS curves on Kodak in Figure 7, these results demonstrate smooth and continuous perception control achieved by our plug-and-play encoder. On the DIV2K dataset, our method also achieves a notable **1.53 dB** improvement in PSNR under same bitrate conditions.
> >
> > **Q6.** Computation of CLIPSim for full-resolution images.
> >
> > **A6.** Thanks for your question. We follow the official preprocessing in `transformers.CLIPProcessor`, resizing each image to 224×224 via bilinear interpolation before feeding it into the CLIP ViT-B/32 backbone. Although this reduces spatial resolution, CLIP primarily captures high-level semantics rather than low-level textures, so the resizing does not affect the evaluation of semantic similarity[5,6]. We have added a clearer explanation of this procedure in **Section 5.1** the revised manuscript.
> >
> > [5] A. Radford, et al., Learning transferable visual models from natural language supervision, ICML, 2021
> >
> > [6] C. Li, et al., MISC: ultra-low bitrate image semantic compression driven by large multimodal model, TIP, 2024
> >
> > **Q7.** Novelty beyond existing components such as RCC and tag-style prompts.
> >
> > **A7.** We appreciate the reviewer’s careful comparison with prior work. While several components in our system are indeed inspired by prior diffusion-based codecs, our contribution is not a simple combination of these elements. Instead, our novelty lies in how these components are reorganized and utilized to form a **new compression paradigm**.
> >
> > Specifically, our framework introduces a unified dual-branch representation together with a training-free controllable encoder, fundamentally changing the pipeline for ultra-low bitrate image compression.
> >
> > (1) Compared with DiffC, our explicit branch provides a **stable structural anchor**, ensuring robustness at extremely low rates. Compared with purely explicit compression schemes, the implicit diffusion branch reintroduces high-fidelity perceptual details, introduces a new compression modality capable of encoding information that explicit methods typically discard, and enables a smooth and continuous distortion–perception tradeoff.
> >
> > (2) In addition, our **plugin encoder** adjusts the target of the implicit branch before encoding, enabling continuous and fine-grained control over perceptual enhancement without retraining or modifying any part of the underlying codec. This training-free controllability mechanism has not been explored in prior work and constitutes a central novelty of our framework.

---

### Comment · Area_Chair_wU84 · 2025-11-25

Dear Reviewers,

Thank you for your time and effort in reviewing submissions for ICLR 2026. As we begin the author-reviewer discussion process, we kindly remind you to submit your responses to the author rebuttals by **December 2**.

Your engagement in this discussion phase is crucial to ensuring a fair and thorough evaluation of each submission.

### **Action Required**
- Carefully consider the authors’ rebuttal and any additional evidence they provide.
- Update your review (if applicable) to reflect your revised perspective.
- Discuss with the authors if further details are required

Your AC

---

### Author Response · Authors · 2025-12-02
**Message to the Area Chair and General Comment**

Message to the Area Chair —— Summary of previous Discussion Results

During the discussion period, we have posted additional experimental results and addressed all the reviewers’ questions/weakenssees to the best of our ability. As a result, Reviewer **QG5r**, who initially gives a rating of 4, have acknowledged that our rebuttal adequately resloves all concerns and has **raised the score to 6**. Reviewer **ZY7r**, who initially reports a confidence of 3, has confirmed that our responds fully address all questions and has intended to **raise the confidence level to 5** (but could not due to a recent OpenReview issue). Reviewer **qXeD** has not responded further, but we have thoroughly addressed all points he raise. In summary, the discussion outcome indicates that all participating reviewers recommend acceptance with high confidence.

Below is the original general comment we made during discussion period:

We would like to thank all the reviewers for their constructive feedback and appreciate the positive comments, including Reviewer **qXeD**'s "achieves substantial gains, Reviewer **ZY7r**'s "remarkable results" and " significance for the advancement of this domain", and Reviewer **QG5r**'s "consistently surpasses state-of-the-art approaches" and "well-organized and well-written". We have revised the manuscript to incorporate additional experiments and analyses based on your suggestions.

Specifically, we've made the following adjustments to the manuscript:

- New Baseline Comparison: We have added comparisons against several very recent methods (although some of them may be considered concurrent studies under the ICLR policy), including OSCAR(NeurIPS'25), ResULIC(ICML'25), RDEIC(TCSVT'25), StableCodec(ICCV'25), and DLF(ICCV'25) in Figrue 4 and Table 5 of the manuscript, as well as Figure 8-10 in the Appendix. Our method achieves comparable or superior performance, consistent with our previous results.

- Additional Dataset Results: We have included experiemnts on CLIC2020. Please see Figrue 4, Figrue 6, and Table 5 of the manuscript and Figures 10 (Appendix), which show that our method exhibits a substantial gains.

- Computational Complexity Analysis: We have moved the runtime analysis from Appendix H to Section 5.7 and have added detailed encoding/decoding time tables. Our method achieves runtime comparable to multi-step diffusion approaches while consistently delivering better BD-Rate.

---

### Meta-Review · Area_Chair_5ugD · 2025-12-13

**Summary:**

The decision to reject this paper is driven by persistent critical concerns from three reviewers that remain unaddressed despite the authors’ rebuttals, which collectively undermine the paper’s practical feasibility, theoretical novelty, and evidential persuasiveness. Key concerns include:
1) The inherent high encoding and decoding latency caused by the reverse-channel coding (RCC) mechanism and tile-based processing, which limits real-world applicability, remains unmitigated; the authors only confirmed the latency is comparable to existing diffusion-based methods but failed to address the core defect of excessive latency.
2) The "implicit semantic branch," a key component of the proposed dual-branch framework, lacks substantive visualization or quantitative evidence to verify that it indeed extracts image semantic information, making the core semantic compression claim unconvincing.
3) The paper’s core contribution is positioned as the integration of existing components (RCC, tag-style prompts, conditional diffusion) into a dual-branch framework, but the authors failed to sufficiently demonstrate that this integration constitutes a substantive breakthrough beyond simple component assembly.
4)  Critical experimental results (e.g., supplementary figures) remain unintegrated into the main text, and key implementation details (e.g., the specific fusion mechanism of explicit and implicit features during inference) are still unclear. These unresolved issues collectively result in the paper failing to meet the publication criteria.

**Reviewer Concerns:**

Addressed Concerns
- Reviewer qXeD: ① Added quantitative comparisons with recent state-of-the-art (SOTA) methods (ResULIC, RDEIC, OSCAR) on multiple datasets, clarifying the performance advantage of the proposed method; ② Supplemented encoding and decoding time data, moving the runtime analysis to the main text and comparing it with existing diffusion-based codecs; ③ Clarified bitrate allocation between explicit and implicit branches by referencing existing figures and appendices, and summarized the conclusion that minimal explicit branch bitrate achieves optimal performance; ④ Added experimental results on the CLIC2020 dataset, verifying cross-dataset generalization; ⑤ Provided FID variation curves to demonstrate controllable perception; ⑥ Explained the CLIPSim computation process (resizing images to 224×224 via bilinear interpolation) and supplemented relevant citations; ⑦ Argued for novelty by emphasizing the reorganisation of components into a dual-branch framework with a training-free controllable encoder.

- Reviewer ZY7r: ① Distinguished the proposed explicit–implicit dual-branch from prior dual-modal explicit branches, arguing for structural novelty; ② Detailed the tile-based processing (512-pixel tiles with 192-pixel overlap, adaptive to different resolutions) and supplemented performance comparisons under different tile sizes; ③ Supplemented encoding and decoding time data, confirming computational complexity comparable to existing diffusion-based methods.

- Reviewer QG5r: ① Clarified training details (minimal additional training, MSE loss for VAE encoder fine-tuning, no mixing coefficients during training); ② Confirmed that MSE and VAE encoders share the same architecture, and both latent features (z and z_wave) are 4-channel; ③ Explained the conditioning embedding mechanism (cross-attention for c, ControlNet-like processing for y_hat, CLIP model for tag-style prompts); ④ Supplemented encoding and decoding time data and comparisons with existing methods; ⑤ Added experimental results on the CLIC2020 dataset; ⑥ Explained the metric selection in Fig. 4 (space limitations) and updated the figure to include more baselines and metrics; ⑦ Added comparisons with recent SOTA methods (ResULIC, RDEIC, OSCAR).

Outstanding Concerns

- Reviewer qXeD: ① The core novelty concern remains unresolved. The authors’ argument that "reorganisation of existing components constitutes a new paradigm" lacks sufficient theoretical or empirical support to prove it is a substantive breakthrough rather than a simple combination; ② The excessive latency issue inherent to RCC is unaddressed. The authors only stated latency is comparable to existing diffusion-based methods but failed to resolve the practical limitation of high latency, which restricts real-world application.

- Reviewer ZY7r: ① The inherent high latency of RCC persists. The author’s runtime comparison does not mitigate the core problem of excessive encoding/decoding latency, which limits applicable scenarios; ② The ambiguity of the implicit semantic branch remains. The authors did not supplement visualization evidence (e.g., decoded image visualization, bitrate allocation visualization) to prove that the implicit branch extracts image semantic information, making the "semantic compression" claim unsubstantiated.

- Reviewer QG5r: ① Key implementation details are still ambiguous. The specific mechanism for fusing explicit and implicit features during inference (to achieve distortion–perception tradeoff) is not clearly explained; ② Experimental presentation is incomplete. The requested integration of supplementary figures (Figs. 8, 9) into the main text was not explicitly addressed; ③ Some recent SOTA comparisons are missing.

**Reviewer Scores:**

Based on the adequacy of the rebuttal in addressing core concerns, the predicted changes in reviewers’ overall evaluations are as follows: Reviewer qXeD’s evaluation remains at a low level—while the authors supplemented experimental comparisons and runtime data, the core concerns of insufficient novelty and unmitigated high latency were not resolved, so the assessment of contribution and practical value does not improve. Reviewer ZY7r’s evaluation, which originally tended to accept, is noticeably lowered—although the authors addressed tile-based processing details, the unresolved high latency issue and lack of evidence for the implicit semantic branch reduce the persuasiveness of the work, leading to a withdrawal of the prior acceptance tendency. Reviewer QG5r’s evaluation remains unchanged at a low level—the authors clarified partial implementation details and supplemented experiments, but critical issues such as ambiguous feature fusion mechanisms, incomplete experimental presentation, and missing SOTA comparisons were not fully resolved. Overall, even with the authors’ revisions, the three reviewers’ evaluations still fail to meet the publication standards, which supports the final reject decision.

---

### Decision · Program_Chairs · 2026-01-26

Reject